# The heteromeric PC-1/PC-2 polycystin complex is activated by the PC-1 N-terminus

Kotdaji Ha[1], Mai Nobuhara[1], Qinzhe Wang[2], Rebecca V Walker[3], Feng Qian[3], Christoph Schartner[1], Erhu Cao[2], Markus Delling[1]*

[1]Department of Physiology, University of California, San Francisco, San Francisco, United States; [2]Department of Biochemistry, University of Utah School of Medicine, Salt Lake City, United States; [3]Division of Nephrology, Department of Medicine, University of Maryland School of Medicine, Baltimore, United States

**Abstract** Mutations in the polycystin proteins, PC-1 and PC-2, result in autosomal dominant polycystic kidney disease (ADPKD) and ultimately renal failure. PC-1 and PC-2 enrich on primary cilia, where they are thought to form a heteromeric ion channel complex. However, a functional understanding of the putative PC-1/PC-2 polycystin complex is lacking due to technical hurdles in reliably measuring its activity. Here we successfully reconstitute the PC-1/PC-2 complex in the plasma membrane of mammalian cells and show that it functions as an outwardly rectifying channel. Using both reconstituted and ciliary polycystin channels, we further show that a soluble fragment generated from the N-terminal extracellular domain of PC-1 functions as an intrinsic agonist that is necessary and sufficient for channel activation. We thus propose that autoproteolytic cleavage of the N-terminus of PC-1, a hotspot for ADPKD mutations, produces a soluble ligand in vivo. These findings establish a mechanistic framework for understanding the role of PC-1/PC-2 heteromers in ADPKD and suggest new therapeutic strategies that would expand upon the limited symptomatic treatments currently available for this progressive, terminal disease.

*For correspondence:
markus.delling@ucsf.edu

Competing interests: The authors declare that no competing interests exist.

## Introduction

The most common monogenetic disease in humans is ADPKD; a major nephropathy characterized by renal cysts that leads to urinary tract infection, hypertension, aneurysm, and ultimately end-stage renal disease (ESRD) (*Kathem et al., 2014*; *Harris and Torres, 2009*). Mutations in one of two ciliary proteins, PC-1 (encoded by *PKD1*) and PC-2 (also known as TRPP1 and encoded by *PKD2*), account for 85% and 15% of ADPKD-causing mutations, respectively (*Harris and Torres, 2009*; *Cornec-Le Gall et al., 2014*).

PC-1 is an 11 transmembrane (TM) protein of 4303 amino acids with a long extracellular N-terminus that contains multiple cell adhesion domains (*Yoder et al., 2002*; *Hughes et al., 1995*). PC-1 and its family members (PC-1L1, PC-1L2, and PC-1L3) share the same 11-TM topology but vary in the size and adhesion domain composition of their N-termini (*Ishimaru et al., 2006*). A unique feature of PC-1 family members, shared only by adhesion G protein-coupled receptors (GPCRs), is the GPCR autoproteolysis inducing (GAIN) domain. This domain catalyzes peptide bond hydrolysis at a GPCR proteolysis site (GPS) that is positioned proximal to the first TM helix (TM1) (*Araç et al., 2012*; *Prömel et al., 2013*; *Trudel et al., 2016*). Autoproteolysis generates two fragments: an N-terminal fragment (NTF) and a multi-TM core (*Qian et al., 2002*). The NTF contains multiple domains, including leucine-rich repeats, a C-type lectin (CTL) domain, immunoglobulin (Ig)-like polycystic kidney disease (PKD) repeat domains, a single low-density lipoprotein (LDL) receptor motif, and the receptor for egg jelly (REJ) domain (*Hughes et al., 1995*; *Babich et al., 2004*; *Moy et al., 1996*).

**eLife digest** On the surface of most animal and other eukaryotic cells are small rod-like protrusions known as primary cilia. Each cilium is encased by a specialized membrane which is enriched in protein complexes that help the cell sense its local environment. Some of these complexes help transport ions in out of the cell, while others act as receptors that receive chemical signals called ligands. A unique ion channel known as the polycystin complex is able to perform both of these roles as it contains a receptor called PC-1 in addition to an ion channel called PC-2.

Various mutations in the genes that code for PC-1 and PC-2 can result in autosomal dominant polycystic kidney disease (ADPKD), which is the most common monogenetic disease in humans. However, due to the small size of primary cilia – which are less than a thousandth of a millimeter thick – little is known about how polycystin complexes are regulated and how mutations lead to ADPKD. To overcome this barrier, Ha et al. modified kidney cells grown in the lab so that PC-1 and PC-2 form a working channel in the plasma membrane which surrounds the entire cell. As the body of a cell is around 10,000 times bigger than the cilium, this allowed the movement of ions across the polycystin complex to be studied using conventional techniques.

Experiments using this newly developed assay revealed that a region at one of the ends of the PC-1 protein, named the C-type lectin domain, is essential for stimulating polycystin complexes. Ha et al. found that this domain of PC-1 is able to cut itself from the protein complex. Further experiments showed that when fragments of PC-1, which contain the C-type lectin domain, are no longer bound to the membrane, they can activate the polycystin channels in cilia as well as the plasma membrane. This suggests that this region of PC-1 may also act as a secreted ligand that can activate other polycystin channels.

Some of the genetic mutations that cause ADPKD likely disrupt the activity of the polycystin complex and reduce its ability to transport ions across the cilia membrane. Therefore, the cell assay created in this study could be used to screen for small molecules that can restore the activity of these ion channels in patients with ADPKD.

The function of the multi-TM core remains poorly defined. Autoproteolysis occurs early in the ER secretory pathway such that the N- terminal fragment remains non-covalently tethered (*Lin et al., 2004*; *Wei et al., 2007*). While mutations within the GPS site of PC-1 result in renal cyst formation, the full functional consequences of autoproteolysis are poorly understood (*Qian et al., 2002*; *Yu et al., 2007*).

Recent studies suggest that PC-1 and PC-2 form a 1:3 heteromer (*Su et al., 2018*; *Yu et al., 2009*). Indeed, the cryo-EM structure of a PC-1/PC-2 heteromer reveals that the 11 transmembrane helices of PC-1 are further divided into two major domains: a peripheral TM1-TM5 complex and a core TM6-TM11 complex that interdigitates with three PC-2 subunits to form a TRP-like ion channel (*Su et al., 2018*). In addition to this PC-1/PC-2 assembly, PC-2 can form homomeric channels, following a classic homotetramer of TRP channels (*Shen et al., 2016*; *Grieben et al., 2017*; *Wilkes et al., 2017*).

Attempts to exogenously express and record from PC-1 and/or PC-2 in whole cell recordings of mammalian cells (*Hanaoka et al., 2000*; *Delmas et al., 2004*; *Cai et al., 2004*) and reconstituted lipid bilayers *González-Perrett et al., 2001* have yielded divergent results, likely due to subcellular enrichment of PC-1 and PC-2 (for a detailed discussion see *Liu et al., 2018*; *Kleene and Kleene, 2017*). In addition, despite recent advances in measuring ion channel activity in small subcellular organelles such as the cilium (*Kleene and Kleene, 2012*; *DeCaen et al., 2013*; *Dong et al., 2008*), it remains technically challenging to study ion channel activity in primary cilia due to their small size ($10^{-15}$ L, approximately the size of a bacterium). This technical hurdle has hindered investigations of the physiological regulation of polycystins and how they lead to renal disease when mutated. The unknown role of PC-1 within the channel complex, in particular, has hindered progress in the field. Some studies suggest that PC-1 acts as a dominant-negative subunit of the heteromeric channel (*Su et al., 2018*), whereas others suggest that it is a pore-forming subunit that fine-tunes ion selectivity (*Wang et al., 2019*). Moreover, PC-1 has been shown to be dispensable for polycystin currents in ciliary membranes (*Liu et al., 2018*). Here we develop cellular assays that allow us to probe the

function of polycystin complexes in the plasma membrane and compare them to polycystin channels in their native primary cilia. Our heterologous system recapitulates the biophysical properties of cilia-localized polycystin channels and demonstrates how distinct PC-1 family members contribute to ion permeation. Importantly, we reveal that the N-terminal domain of PC-1 functions as a soluble ligand and is an indispensable activator of the polycystin complex, providing further evidence for chemosensory regulation of the polycystin complex (*Delling et al., 2016*).

## Results

### PC-1 and PC-2 form functional channels in the plasma membrane

Because robust assays to investigate the heteromeric polycystin complex are lacking, we developed an assay system to determine the contribution of PC-1 family members to polycystin function. We were able to target PC-1 to the plasma membrane of HEK293 cells (*Figure 1B*) and mIMCD3 cells (*Figure 1—figure supplement 1E*) by replacing its endogenous signal peptide with a strong Ig κ-chain secretion sequence followed by an HA tag (*s*PC-1; *Figure 1A*, *Table 1*; *Salehi-Najafabadi et al., 2017*; *Zhu et al., 2011*). The HA tag allowed us to quantify *s*PC-1 plasma membrane expression by measuring surface staining following addition of anti-HA antibodies to non-permeabilized cells. Co-expression of *s*PC-1 and PC-2 resulted in strong HA surface staining, whereas expression of *s*PC-1 alone resulted in only marginal surface staining, in both HEK293 cells (*Figure 1B and C*) and IMCD3 cells (*Figure 1—figure supplement 1D and E*). Co-expression of PC-2 and PC-1 with an extracellular HA tag but without the endogenous leader sequence (HAPC-1) failed to generate comparable HA surface staining, supporting the notion that an Ig κ-secretion sequence promotes membrane localization of the complex (*Figure 1B*). To test whether PC-2 facilitates *s*PC-1 membrane localization by promoting ER exit or by co-migration to the plasma membrane, we inserted a FLAG epitope into the extracellular tetragonal opening for polycystins (TOP) domain of PC-2 (PC-2$_{FLAG}$) (*Figure 1—figure supplement 1A*, *Table 1*). Robust surface HA and FLAG staining only occurred when both *s*PC-1 and PC-2$_{FLAG}$ were co-expressed, suggesting that they are shuttled simultaneously to the plasma membrane (*Figure 1—figure supplement 1B and C*). Similarly, co-expression of *s*PC-1 with PC-2 containing a recently in an oocyte expression system characterized gain of function mutation of PC-2 (PC-2$_{F604P}$, *Table 1*; *Arif Pavel et al., 2016*) resulted in comparable HA surface expression (*Figure 1B and C*).

Having established robust plasma membrane expression of *s*PC-1/PC-2 and *s*PC-1/PC-2$_{F604P}$, we asked whether these proteins form functional channels using whole cell patch clamp recordings. Interestingly, only *s*PC-1/PC-2$_{F604P}$ produced constitutively active, outward rectifying currents (122.6 ± 23.0 pA/pF at +100 mV, n = 7) with a half-maximal activation voltage of +112.5 mV (*Figure 1D, E and G*). *s*PC-1/PC-2 (5.1 ± 1.0 pA/pF, n = 11), HAPC-1/PC-2$_{F604P}$ (5.0 ± 2.3 pA/pF, n = 15) and PC-2$_{F604P}$ alone (6.4 ± 2.1 pA/pF, n = 12) yielded negligible currents at +100 mV (*Figure 1F*). To investigate the functional contribution of PC-1 to heteromeric polycystin complexes, we mutated three positively charged amino acids in the putative ion permeation pathway to the small, uncharged amino acid glycine (R4100G, R4107G, and H4111G) in order to facilitate cation permeation (*s*PC-1$_{pore}$; *Figure 1H*, *Table 1*). In combination with PC-2$_{F604P}$, currents of 71.1 ± 5.4 pA/pF (n = 6) were generated at +180 mV (*s*PC-1$_{pore}$/PC-2$_{F604P}$; *Figure 1I*). Although the recorded current density was ~50 pA/pF smaller than for *s*PC-1/PC-2$_{F604P}$ (*Figure 1J*), the relative open probability at −100 mV was higher (0.3 ± 0.0 for *s*PC-1$_{pore}$/PC-2$_{F604P}$; 0.1 ± 0.0 for *s*PC-1/PC-2$_{F604P}$), suggesting that the pore mutant resides in a more open state (*Figure 1K*).

Collectively, these results demonstrate that PC-1/PC-2 subunits form a functional channel when reconstituted in mammalian cells and support the notion that PC-1 subunits form part of the core channel complex.

### The PC-1 N-terminus is essential for polycystin complex activation

We sought to determine whether PC-1 subunits are involved in the regulation of the polycystin complex by investigating the functional role of the N-terminal domain. We reasoned that the N-terminus must be a critical determinant of polycystin function because ADPKD-causing mutations occur with high frequency in this region and therefore investigated a family member with a divergent N-terminus, PC-1L3 (*Figure 2A* and *Figure 2—figure supplement 1A*). Substitution of PC-1L3's signal

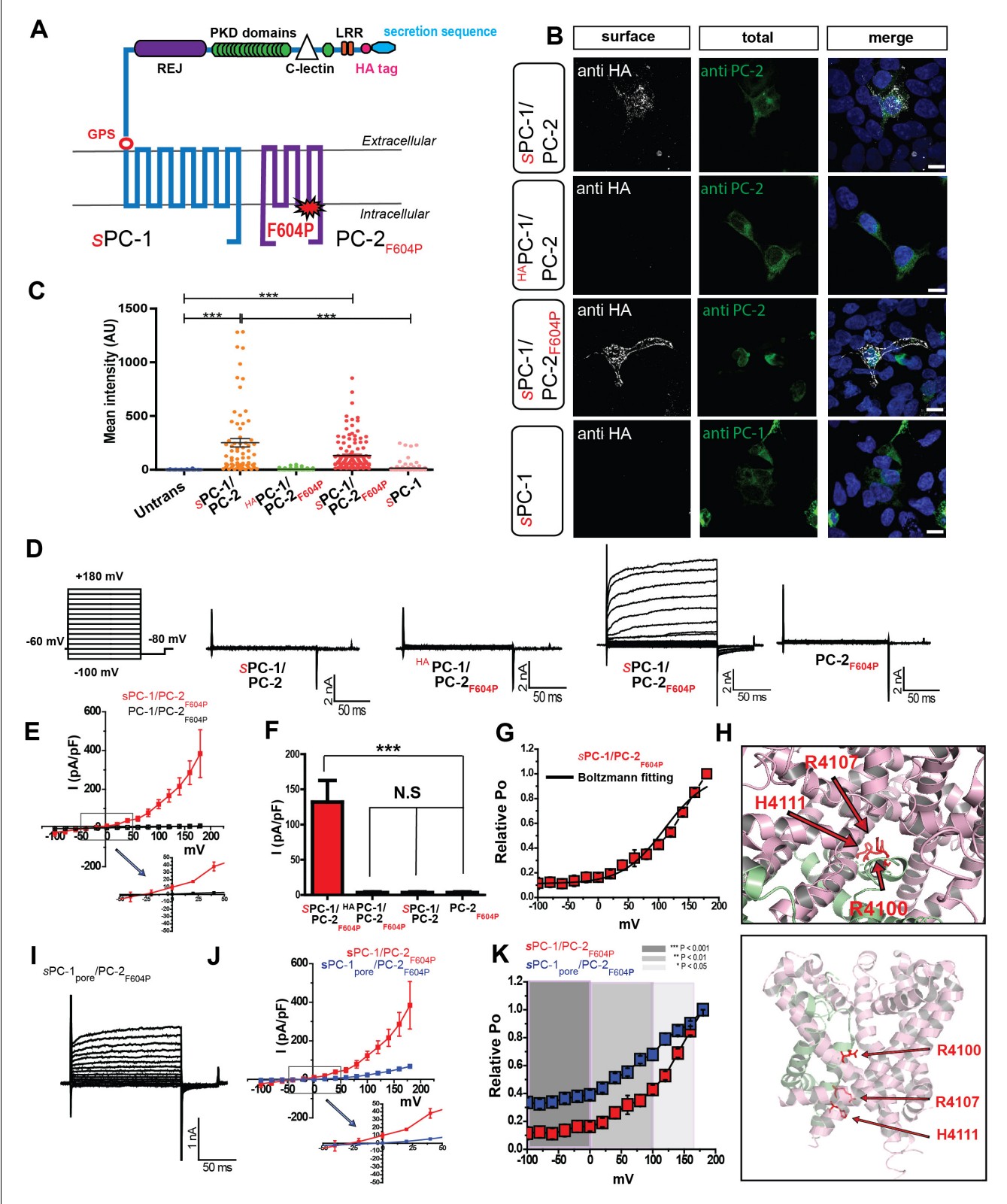

**Figure 1.** PC-1 and PC-2 form functional channels in the plasma membrane. (**A**) Illustration of PC-1 and PC-2$_{F604P}$ topology highlighting extracellular HA and secretion sequence (indicated with an *s* prefix) in PC-1. (**B**) Expression and immunostaining of HEK293 cells transfected with *s*PC-1/PC-2, $^{HA}$PC-1/PC-2 with endogenous leader sequence, *s*PC-1/PC-2$_{F604P}$, and *s*PC-1 alone. Left column, representative images of HA surface staining (see methods). Middle column, total staining for PC-2 or PC-1. Right column, merged images. Scale bar: 10 µm. (**C**) Quantification of relative anti-HA surface staining

*Figure 1 continued on next page*

*Figure 1 continued*

shown in B. ***: p<0.0001. (D) Representative currents from each construct recorded in the whole cell patch clamp configuration in response to voltage step pulses (left). (E) Whole-cell I-V relationship of $sPC$-1/PC-$2_{F604P}$ (n = 6, red) and PC-1/PC-$2_{F604P}$ (n = 10, white). Arrow indicates the expanded view of I-V relationship between membrane potentials of −50 mV and +50 mV. (F) Current densities for $sPC$-1/PC-$2_{F604P}$, $^{HA}$PC-1/PC-$2_{F604P}$, $sPC$-1/PC-2, and PC-$2_{F604P}$ alone at +100 mV. (G) Relative open probability of $sPC$-1/PC-$2_{F604P}$ at −80 mV calculated using a Boltzmann distribution fitted to I/Imin of tail current amplitudes, black line (n = 6, red). (H) Top view of the human PC-1/PC-2 polycystin complex using pymol (*Su et al., 2018*). Positively charged R4100, R4107, and H4111 are highlighted in red and displayed in stick shape (top). Side view of selectivity filter showing the location of three positively charged amino acids on PC-1 TM-11 (bottom). (I) Representative whole cell currents of $sPC$-$1_{pore}$/PC-$2_{F604P}$. (J) I-V relationship of $sPC$-1/PC-$2_{F604P}$ (n = 6, red) and $sPC$-$1_{pore}$/PC-$2_{F604P}$ (n = 6, blue). Arrow indicates the expanded view of I-V relationship between −50 mV and +50 mV. Current densities of $sPC$-1/PC-$2_{F604P}$ and $sPC$-$1_{pore}$/PC-$2_{F604P}$ were compared at +100 mV holding potential. (K) Relative open probability analysis of $sPC$-1/PC-$2_{F604P}$ (n = 6, red) and $sPC$-$1_{pore}$/PC-$2_{F604P}$ (n = 6, blue) during a −80 mV tail pulse. Statistical significance is indicated by the background. ***p<0.001 (dark gray), **p<0.01 (gray), *p<0.05 (light gray). Statistical analysis was computed using student t-test unpaired one-pair.

The online version of this article includes the following figure supplement(s) for figure 1:

**Figure supplement 1.** Surface expression of PC-1, PC-$2_{FLAG}$, and PC-1L3 in HEK293 and mIMCD3 cells.

peptide with the κ IgG secretion sequence ($sPC$-1L3, *Table 1*) resulted in strong surface HA staining when co-expressed with PC-2 or PC-$2_{F604P}$ (*Figure 2B and D*). However, we were unable to record convincing currents in $sPC$-1L3/PC-$2_{F604P}$-transfected HEK cells (3.5 ± 0.0 pA/pF, n = 12), suggesting that, although present in the plasma membrane, the complex is inactive (*Figure 2C*). To determine the structural domains in PC-1 subunits that are required for channel activation, we generated

**Table 1.** Constructs used in this study.

| Construct name | Protein tag | Mutation |
|---|---|---|
| $sPC$-1/PC-$2_{F604P}$ | κ-IgG signal peptide and HA tag at 5′ end of PC-1 | F604P in PC-2 |
| PC-1/PC-$2_{F604P}$ | HA tag at 5′ end of PC-1 | F604P in PC-2 |
| $sPC$-1/PC-2 | κ-IgG signal peptide and HA tag at 5′ end of PC-1 | wild type PC-2 |
| $sPC$-1/PC-$2_{FLAG}$ | κ-IgG signal peptide and HA tag at 5′ end of PC-1, FLAG insertion in PC-2 TOP domain (pos. 904):...PSNGT-DYKDDDK-SFIFY... | |
| PC-$2_{F604P}$ | - | F604P in PC-2 |
| $sPC$-$1_{pore}$/PC-$2_{F604P}$ | κ-IgG signal peptide and HA tag at 5′ end of PC-1 | R4100G, R4107G, H4111G in PC-1 F604P in PC-2 |
| PC-$1^{ΔNT}$/PC-2 | HA tag (P2Y12) | Substitution of PC-1 N-terminus adjacent to TM-1 (position 9184) with P2Y12 N-terminus. (HA_P2Y12_mypydvpdyaqavdnltsapgntslctrdykitq)-(vrfvfp..._PC-1) |
| PC-$1^{ΔNT}$/PC-$2_{F604P}$ | HA tag (P2Y12) | Substitution of PC-1 N-terminus with P2Y12 N-terminus F604P in PC-2 |
| $sPC$-1L3/PC-2 | κ-IgG signal peptide and HA tag at 5′ end of mPC-1L3 | - |
| $sPC$-1L3$^{1NT}$/PC-2 | κ-IgG signal peptide and HA tag at 5′ end of PC-1 N-terminus | Substitution of mPC-1L3 N-terminus with hPC-1 N-terminus at GPS cleavage (hPC-1...CLTR-HLTFFSS...mPC-1L3) |
| $sPC$-$1^{1L3NT}$/PC-2 | κ-IgG signal peptide and HA tag at 5′ end of PC-1L3 N-terminus | Substitution of hPC-1 N-terminus with 1-1052aa of mPC-1L3 at GPS cleavage (mPC-1L3... QCLCD-HLTAFGA...hPC-1) |
| $sPC$-$1^{1L3CTL}$/PC-2F604P | κ-IgG signal peptide and HA tag at 5′ end of PC-1 | Replacement of PC-1 C-type lectin domain (407-547aa) with C-type lectin domain of PC-1L3 (30-150aa) |
| $sPC$-1L3$^{1CTL}$/PC-2F604P | κ-IgG signal peptide and HA tag at 5′ end of PC-1L3 | Replacement of PC-1L3 C-type lectin domain (30-138aa) with C-type lectin domain of PC-1 (415-535aa) |
| $sPC$-1/PC-$2^{N375Q}_{F604P}$ | κ-IgG signal peptide and HA tag at 5′ end of PC-1 N-terminus | F604P in PC-2, N375Q in PC-2 |

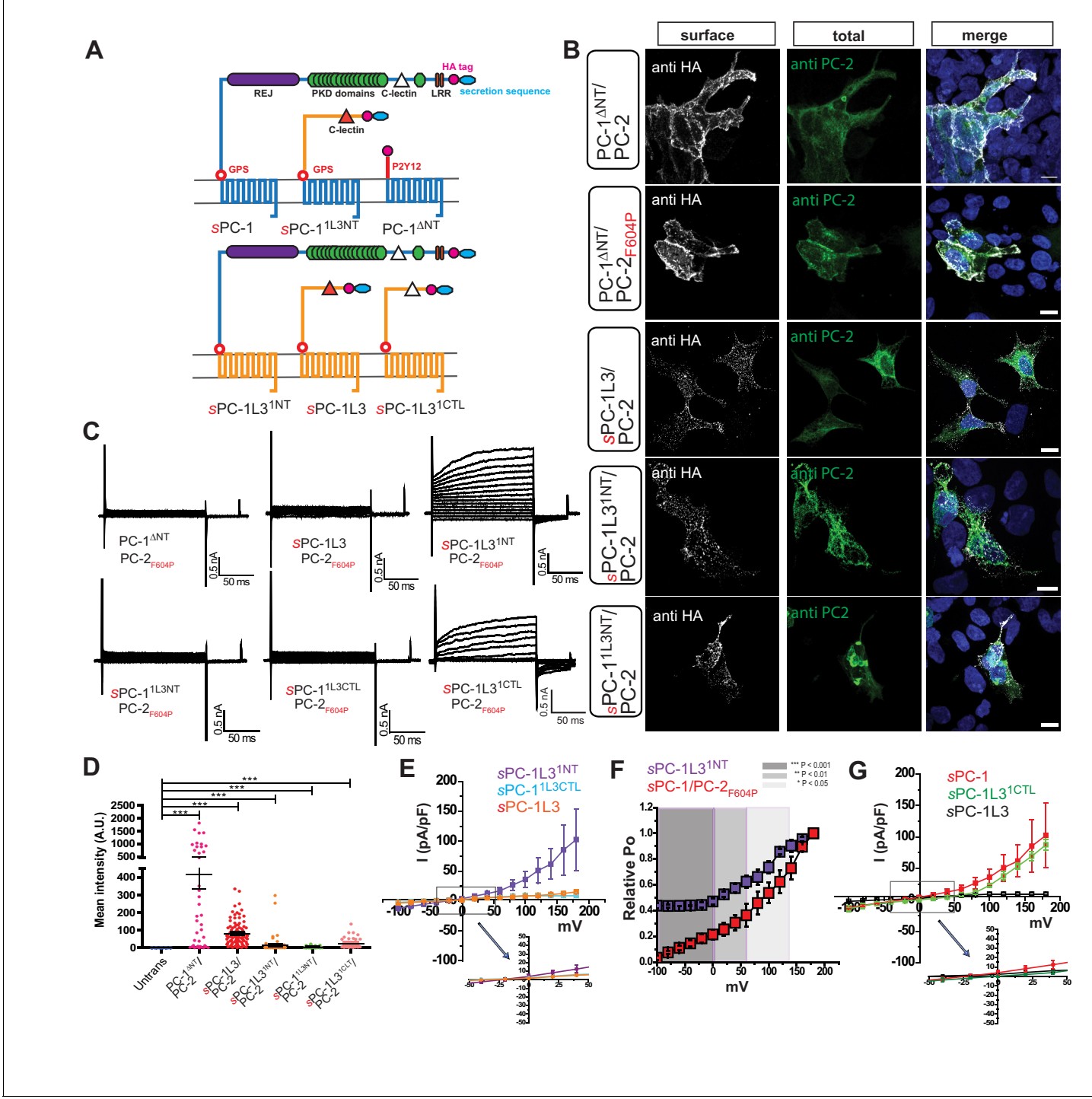

**Figure 2.** The PC-1 N-terminus is essential for polycystin complex activation. (**A**) Schematic diagram illustrating topology of PC-1 (blue) and PC-1L3 (yellow), including PC-1$^{\Delta NT}$ (top), PC-1 N-terminal chimeras (top), and PC-1L3 N-terminal chimeras (bottom). (**B**) Immunostaining of HEK293 cells transfected with indicated chimera. Left, representative images of anti-HA surface staining. Middle, total staining for PC-2. Right, merged images. Scale bar: 10 μm. (**C**) Representative whole cell current traces of each construct obtained from the voltage step pulses shown in *Figure 1D*. (**D**) Quantification of anti-HA surface staining of each construct shown in (**B**). ***p<0.001. (**E**) I-V relationships for N-terminal chimeras $s$PC-1L3$^{1NT}$/PC-2$_{F604P}$ (n = 5, purple), $s$PC-1$^{1L3CTL}$/PC-2$_{F604P}$ (n = 10, light blue) and $s$PC-1L3/PC-2$_{F604P}$ (n = 12, orange). (**F**) Relative open probabilities for $s$PC-1L3$^{1NT}$/PC-2$_{F604P}$ (n = 5, purple) and $s$PC-1/PC-2$_{F604P}$ (n = 6, red, dataset from *Figure 1E*) obtained during tail pulses at −80 mV. Statistical significance is indicated by the background. ***p<0.001 (dark gray), **p<0.01 (gray), *p<0.05 (light gray). Statistical analysis was computed using student t-test unpaired one-pair. (**G**) I-V relationships for $s$PC-1/PC-2$_{F604P}$ (n = 6, red), $s$PC-1L3$^{1CTL}$/PC-2$_{F604P}$ (n = 5, green), and $s$PC-1L3/PC-2$_{F604P}$ (n = 12, black).

*Figure 2 continued on next page*

*Figure 2 continued*

The online version of this article includes the following figure supplement(s) for figure 2:

**Figure supplement 1.** Protein purification of PC-1 N-terminal fragments.

chimeras between PC-1 and PC-1L3: the N-terminus of PC-1 fused to the 11-TM core of PC-1L3 preceding the GPS cleavage site (sPC-1L3$^{1NT}$) and the opposing arrangement (sPC-1$^{1L3NT}$) (***Figure 2A***, ***Table 1***). We also substituted the entire PC-1 N-terminus with the 26 amino acid-long N-terminus of the P2Y12 receptor and an extracellular HA tag (PC-1$^{\Delta NT}$, ***Table 1***); a modification known to increase surface expression of GPCRs without impairing functionality (***Liberles and Buck, 2006***; ***Liebscher et al., 2015***).

All chimeras localized to the plasma membrane when co-expressed with PC-2$_{F604P}$ (***Figure 2B and D***). PC-1$^{\Delta NT}$ resulted in the strongest surface staining, likely due to the smaller size of this protein. Of the three chimeras, only sPC-1L3$^{1NT}$/PC-2$_{F604P}$ generated small but appreciable outward currents (35.9 ± 12.9 pA/pF, n = 5). Currents in cells expressing sPC-1$^{1L3NT}$/PC-2$_{F604P}$ (9.1 ± 1.9 pA/pF, n = 10) and PC-1$^{\Delta NT}$/PC-2$_{F604P}$ (7.6 ± 1.5 pA/pF, n = 10) were negligible (***Figure 2C and E***). Furthermore, the relative open probability of the sPC-1L3$^{1NT}$/PC-2$_{F604P}$ chimera at −100 mV (0.44 ± 0.0) was ~four-fold higher than sPC-1/PC-2$_{F604P}$ (0.1 ± 0.0) and similar to that for sPC-1$_{pore}$/PC-2$_{F604P}$ (0.3 ± 0.0) (***Figure 2F***). Interestingly, the predicted ion permeation region of PC-1L3 does not contain the positively charged amino acids found in PC-1, offering a possible explanation for the greater open probability observed in the sPC-1L3$^{1NT}$/PC-2$_{F604P}$ chimeras (***Figure 2—figure supplement 1A***). In terms of channel regulation, these results support the hypothesis that the N-terminus of PC-1 is a key determinant of polycystin complex activation.

## Polycystin activation depends on CTL and TOP domains

To gain a more detailed understanding of the role played by the N-terminus in polycystin complex activation, we replaced smaller regions of the PC-1L3 N-terminus with those from PC-1. One chimera replaced the 108 AA CTL domain of PC-1L3 with the CTL of PC-1 (sPC-1L3$^{1CTL}$; ***Figure 2A***, ***Table 1***). sPC-1L3$^{1CTL}$/PC-2$_{F604P}$ generated robust outward currents (87.5 ± 9.0 pA/pF, n = 5) (***Figure 2C and G***) whereas the inverse chimera (sPC-1$^{1L3CTL}$) produced only minimal currents (9.1 ± 1.9 pA/pF, n = 10). These data strongly suggest that the PC-1 CTL domain is essential to activate the polycystin complex.

In mammalian cells, the PC-1 ectodomain undergoes at least one proteolytic cleavage event at the GPS site, suggesting that the N-terminus might be liberated as a secreted ligand. Indeed, it has been reported that PC-1 N-terminus-containing exosomes interact with primary cilia in kidney nephrons (***Hogan et al., 2009***). We therefore investigated whether fragments of the PC-1 N-terminus can regulate the polycystin complex. We purified various PC-1 NTFs from HEK cell supernatant (***Figure 2—figure supplement 1C–1E***) and tested their ability to confer channel activity on PC-1$^{\Delta NT}$/PC-2$_{F604P}$ heteromers when included in the bath solution. Surprisingly, application of an NTF spanning the distal region AA 24–852 (NTF$^{24-852}$), including the CTL domain between PKD I and II (amino acids 415–531; ***Figure 2—figure supplement 1B and C***), conferred the ability to carry outward currents on PC-1$^{\Delta NT}$/PC-2$_{F604P}$ heteromers (65.5 ± 37.2 pA/pF, n = 6) (***Figure 3A and C***). Similarly, a shorter NTF containing the CTL domain (NTF$^{263-535}$; ***Figure 2—figure supplement 1B and C***) was also able to induce outward currents (61.4 ± 10.5 pA/pF, n = 4; ***Figures 3A, B and C***). In both cases, the currents were smaller than those for sPC-1/PC-2$_{F604P}$ (***Figure 1F***). Boiling NTF$^{263-535}$ at 95°C for 10 min rendered it inactive (***Figure 3B***). Next, we perfused HEK cells overexpressing PC-1$^{\Delta NT}$/PC-2$_{F604P}$ heteromers with NTF$^{263-535}$ and measured acute changes in current density. While PC-1$^{\Delta NT}$/PC-2$_{F604P}$ channels remained inactive in the absence of NTF$^{263-535}$, acute application increased current density by 765.1 ± 185.05% (from 5.9 ± 1.3 pA/pF to 59.1 ± 19.3 pA/pF, n = 7) within 10 s of application (***Figures 3D, F and G***). However, non-transfected cells did not respond to NTF$^{263-535}$ under comparable condition (4.6 ± 1.6 pA/pF to 6.8 ± 1.4 pA/pF, n = 9) (***Figure 3E and G***). We then asked whether the removal of PC-1 N-terminus inactivates the polycystin heteromer. To test this possibility, we perfused sPC-1/PC-2$_{F604P}$ overexpressing HEK cells with 0.125% trypsin, assuming that proteolytic digestion of extracellular proteins also removes critical regions within the PC-1 N-terminus. As shown in ***Figure 3H*** perfusion with 0.125% trypsin decreased sPC-1/PC-2$_{F604P}$

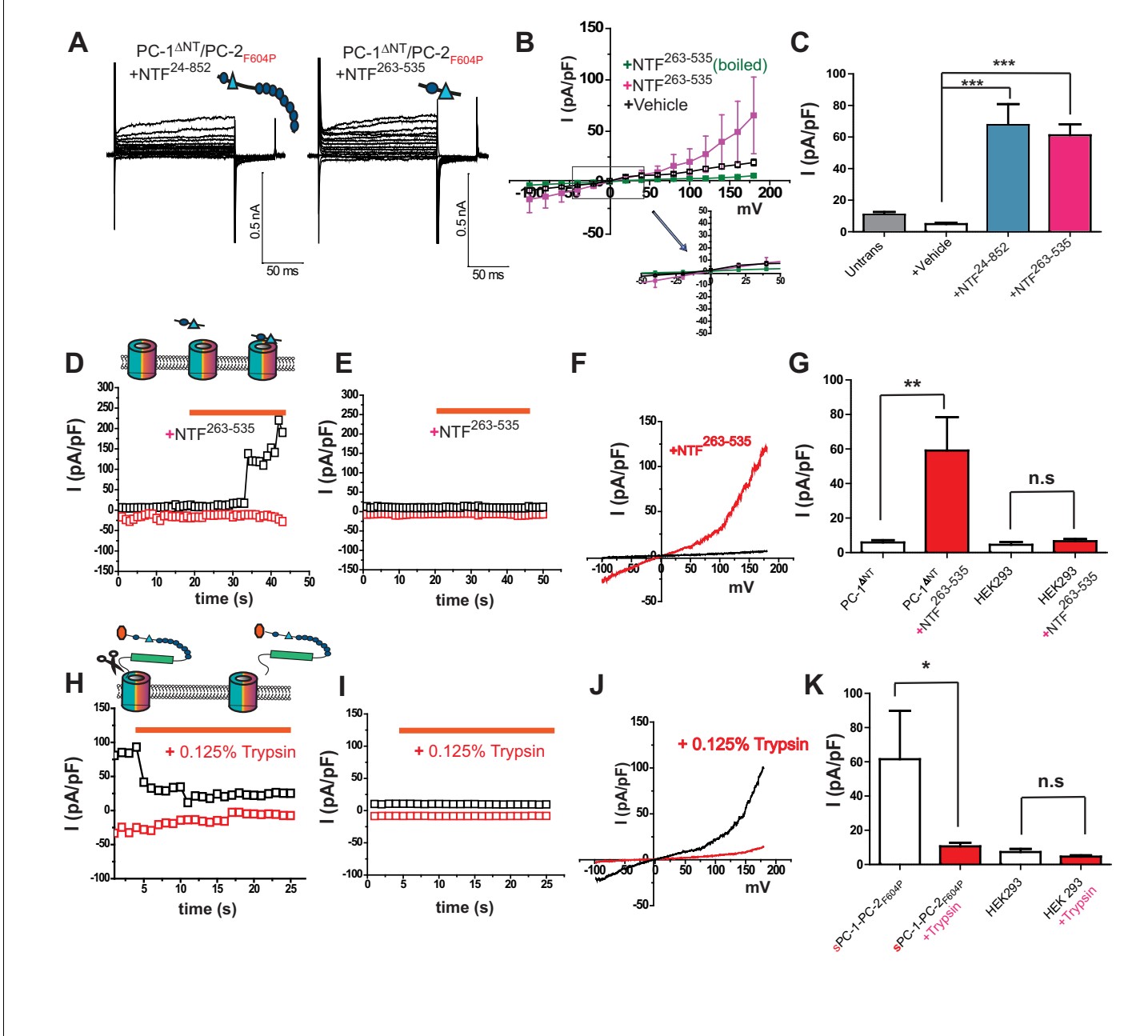

**Figure 3.** Polycystin activation depends on the C-Type lectin domain. (**A**) Representative whole cell currents of PC-1$^{\Delta NT}$/PC-2$_{F604P}$ after application of PC-1 NTF$^{24-852}$ (left) or NTF$^{263-535}$ (right). (**B**) I-V relationships of PC-1$^{\Delta NT}$/PC-2$_{F604P}$ with addition of NTF$^{263-535}$ (n = 6, pink), PC-1$^{\Delta NT}$/PC-2$_{F604P}$ with addition of heat-inactivated fragment (n = 8, black), and PC1$^{\Delta NT}$/PC-2$_{F604P}$ alone (n = 8, green). (**C**) Current densities of untransfected cells (gray) and PC-1$^{\Delta NT}$/PC-2$_{F604P}$ after addition of control medium (n = 7, white), NTF$^{24-852}$ (n = 5, blue) and NTF$^{263-535}$, (n = 6, pink). ***, p<0.001. (**D**) Time course of PC-1$^{\Delta NT}$/PC-2$_{F604P}$ transfected cells in response to NTF$^{263-535}$. Currents are recorded at +180 mV (black) and −100 mV (red). Orange bar indicates addition of NTF. (**E**) Time course of untransfected cells in response to NTF$^{263-535}$. Conditions are identical to D. (**F**) I-V relationships obtained by voltage ramp pulse for PC-1$^{\Delta NT}$/PC-2$_{F604P}$ before (black) or after (red) NTF$^{263-535}$ application. (**G**) Current densities at +100 mV potential from PC-1$^{\Delta NT}$/PC-2$_{F604P}$ (n = 7) or untransfected cells (n = 9) before (blank) or after application (red) of NTF$^{263-535}$. (**H**) Time course of sPC-1/PC-2$_{F604P}$ transfected cells in response to application of 0.125% trypsin. Currents are recorded at +180 mV (black) and −100 mV (red). Orange bar indicates addition of trypsin. (**J**) I-V relationships for sPC-1/PC-2$_{F604P}$ before (black) and after (red) 0.125% trypsin application. (**K**) Current densities at +100 mV potential from for sPC-1/PC-2$_{F604P}$ (n = 6) and untransfected cells (n = 5) before (blank) or after (red) 0.125% trypsin application.

dependent currents by 78.4 ± 4.9% (from 52.0 ± 17.1 pA/pF to 8.9 ± 1.4 pA/pF, n = 6, *Figure 3H and J*, and K). By contrast, untransfected cells did not respond to 0.125% trypsin (7.2 ± 1.9 pA/pF versus 4.6 ± 0.8 pA/pF, n = 5). Taken together, these data suggest that the N-terminus including the CTL of PC-1 is necessary and sufficient to activate the polycystin complex.

Next, we asked how the extracellular CTL domain in PC-1 might regulate activation of the entire channel complex. We reasoned that the TOP domain in PC-2 is well positioned to function as a surface receptor as it forms a bridge between the pore domain and voltage sensor-like domain, and is critical for channel activation (*Shen et al., 2016*; *Vien et al., 2020*). In addition, different glycosylation patterns of the TOP domain might underlie conformational changes in the homomeric PC-2 complex (*Wilkes et al., 2017*). We observed prominent glycosylation of the TOP domain in PC-2 at position N375, N362, and N328 (*Figure 4A*). Moreover, although mutation of N375 to glutamine did not affect surface localization of $s$PC-1/PC-2$_{F604P}^{N375Q}$ (*Figure 4B and C*), it rendered the channel inactive (*Figure 4D and E*). Thus, glycosylation of the TOP domain in PC-2 appears to be required for CTL-dependent polycystin activation.

## N-terminal PC-1 fragments activate ciliary polycystin complexes

Although the results so far strongly suggest that the PC-1 CTL domain is critical for activating heterologous polycystin complexes, native polycystin complexes reside in the specialized membranes of

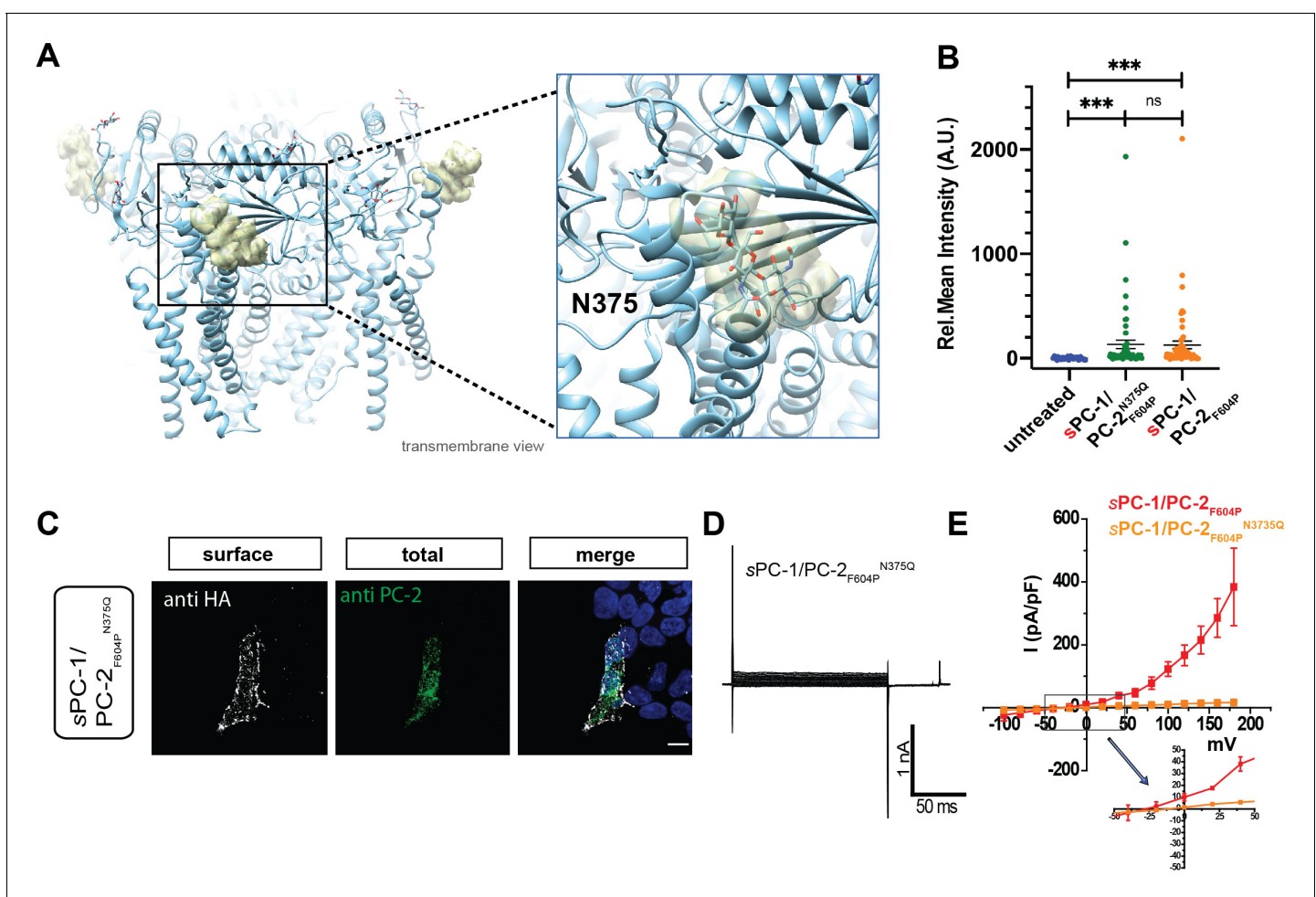

**Figure 4.** C-type lectin binds to the TOP domain of PC-2. (A) Location of carbohydrates (yellow mesh) bound to N375 in the TOP domain of the PC-2 channel (protein data bank ID: 5T4D). Right, expanded view of carbohydrate binding to N375 site. (B) Quantification of anti-HA surface staining of $s$PC-1/PC-2$_{F604P}$ and $s$PC-1/PC-2$_{F604P}^{N375Q}$ compared to untransfected cells. ***p<0.001. (C) Immunostaining of HEK293 cells transfected with $s$PC-1/PC-2$_{F604P}^{N375Q}$. Scale bar: 10 μm. (D) Representative whole cell current traces of $s$PC-1/PC-2$_{F604P}^{N375Q}$. (E) I-V relationships for $s$PC-1/PC-2$_{F604P}^{N375Q}$ (n = 11, orange) and $s$PC-1/PC-2$_{F604P}$ (n = 6, red, dataset from *Figure 1E*).

primary cilia (*Liu et al., 2018*; *Kleene and Kleene, 2017*; *Vien et al., 2020*). We, therefore, investigated whether the PC-1 N-terminus is able to activate polycystin complexes in the cilia of polarized mouse epithelial (mIMCD-3) cells using excised ciliary membrane patch clamp recordings (*Figure 5—figure supplement 1B*). Without any NTFs, we readily detected single channel openings at potentials more positive than +60 mV and more negative than −80 mV (*Figure 5A*) in 7 out of 38 cilia. After ablating PC-1 expression in mIMCD3 cells using CRISPR/CAS9 (*Figure 5—figure supplement 1A*) ciliary channels remained readily detectable in 4 of 13 cilia (*Figure 5B*), in agreement with previous reports (*Liu et al., 2018*). In addition, single channel conductance was comparable between wt mIMCD3 ($\gamma$ = 79.5 ± 0.1 pS) and PC-1 knockout cilia ($\gamma$ = 82.3 ± 0.0 pS) (*Figure 5—figure supplements 1G, H, 2A and B*). However, we noted that the open probability of channels at negative membrane potentials in PC-1 knock out primary cilia (0.58 ± 0.01) was higher than in wt IMCD3 cilia (0.08 ± 0.01) (*Figure 5G*). These data suggest that PC-2 forms a homomeric complex in the absence of PC-1 subunits with a higher open probability than that of PC-1/PC-2 heteromers. The results also support the idea that PC-1 subunits impede cation flux across the membrane, in agreement with our PC-1 pore mutant data (*Figure 1I–K*). Ablation of PC-2 expression using CRISPR/CAS9 rendered primary cilia electrically silent (n = 11), confirming that PC-2 subunits are an essential component of polycystin channels (*Figure 5C*; *Liu et al., 2018*; *Kleene and Kleene, 2017*).

To test whether NTFs can activate the native polycystin complex, we recorded single channels in inside out patches pulled from primary cilia in the presence of 0.7 µg/mL (equal to ~50 nM final concentration) of NTF$^{263-535}$ in the recording pipette. Although the percentage of electrically active cilia (n = 4/13) remained largely unaffected, NTF$^{263-535}$ increased the probability of single channels openings by ~five-fold and significantly prolonged the duration of channel openings at negative membrane potentials (*Figure 5E–G*).

Collectively, these findings demonstrate that PC-2 subunits are essential for functional polycystin channels in primary cilia, that PC-1 subunits also participate in the native heteromeric complex, and that NTF$^{263-535}$ can activate this complex under physiological conditions.

## NTF$^{263-535}$ activates Ca$^{2+}$ influx into primary cilia

Having established that NTF$^{263-535}$ can activate ciliary polycystin channels, we asked whether it could also trigger Ca$^{2+}$ entry through the channel. We initially attempted single channel recordings from mIMCD-3 wt cilia under conditions in which Ca$^{2+}$ is the sole charge carrier (see Materials and methods). We were unable to measure currents, even at very negative membrane potentials (−140 mV) (*Figure 6A*), in agreement with previously published results (*Liu et al., 2018*). Surprisingly, however, the application of NTF$^{263-535}$ revealed channel openings close to physiological membrane potentials (*Figure 6B*). Single channel conductance was smaller when using Ca$^{2+}$ instead of Na$^{+}$ and K$^{+}$ as the charge carrier (47.5 pS, *Figure 6C*). Voltage-dependent open probability and open time remained unchanged (*Figure 6D and E*).

We subsequently tested whether NTF$^{263-535}$ can also induce changes in ciliary Ca$^{2+}$ concentration ([Ca$^{2+}$]$_{cilium}$) using an improvement to our recently-published ciliary ratiometric Ca$^{2+}$ sensors (*Delling et al., 2016*; *Delling et al., 2013*). We fused mScarlet and GCaMP6s to Arl13b (Arl13b-sca-G6) and stably expressed the construct in both wt and PC-2 knockout mIMCD-3 cells. Cells growing on the bottom side of transwell inserts were imaged using TIRF microscopy (TIRFM) to detect localized changes in [Ca$^{2+}$]$_{cilium}$ (*Ishikawa and Marshall, 2015*). Application of NTF$^{263-535}$ generated multiple rapid spike-like elevations of [Ca$^{2+}$]$_{cilium}$ ($\Delta$F/F > 0.35, n = 35 cilia) whereas untreated cells (n = 22) and cells treated with boiled NTF$^{263-535}$ (n = 28) did not exhibit such fluctuations (*Figure 6F and G*). In addition, NTF$^{263-535}$ did not elicit [Ca$^{2+}$]$_{cilium}$ fluctuations in PC-2 knockout cells (n = 15; *Figure 6H and I*). These results show that polycystin channels conduct Ca$^{2+}$ when activated by NTF$^{263-535}$.

## Discussion

We have demonstrated that PC-1 and related family members form a functional ion channel with PC-2. These heteromeric complexes exhibit distinct biophysical properties, suggesting that PC-1 members are critical for adjusting the electrical excitability of primary cilia in response to the local environment. By expressing membrane-targeted PC-1 or PC-1L3 with PC-2 in mammalian cells, we show that the PC-1 N-terminus is essential for the activation of complexes containing PC-2 subunits,

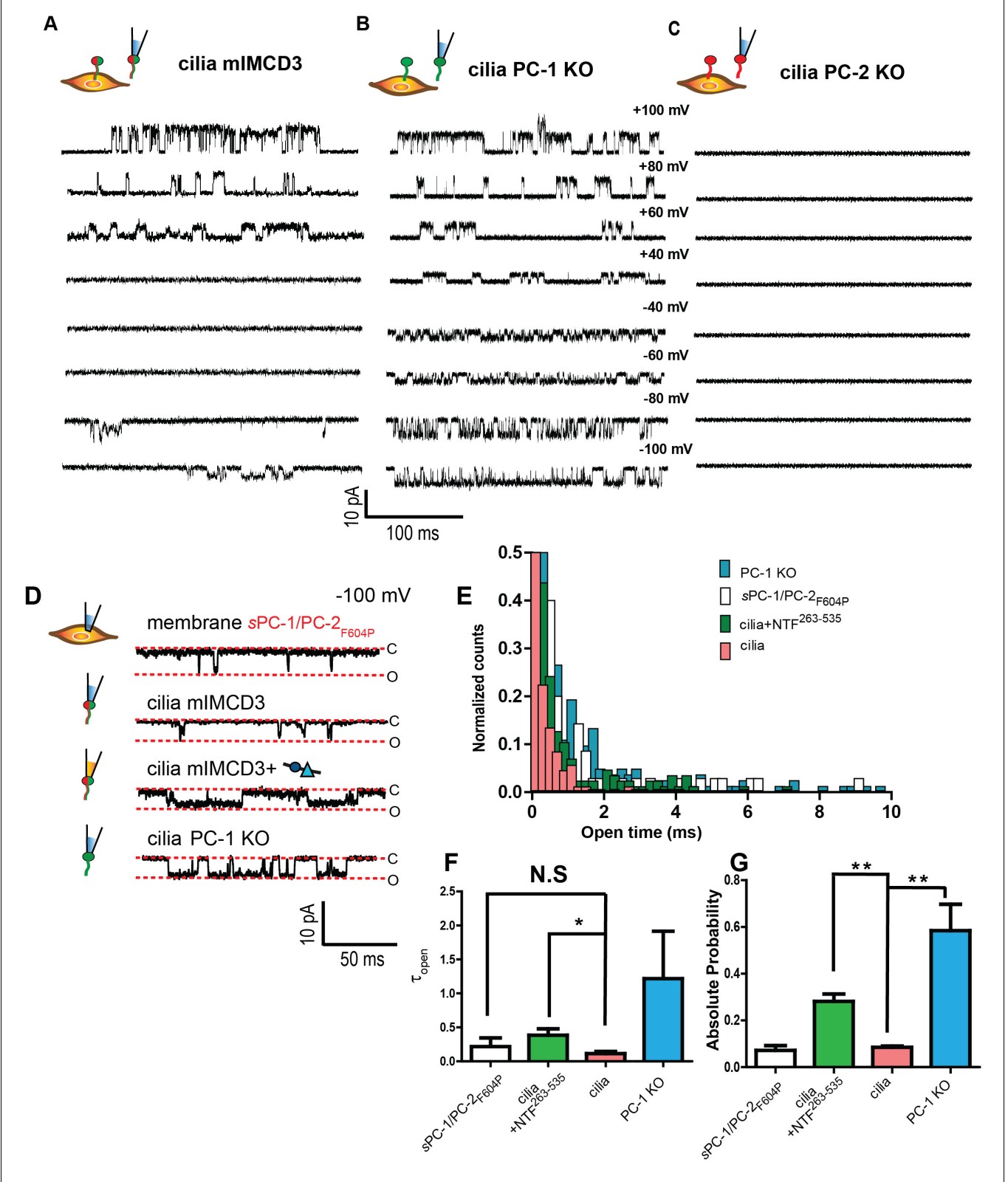

**Figure 5.** An N-terminal PC-1 fragment activates ciliary polycystin complexes. (A, B, C) Excised ciliary inside-out patch clamp recordings of channels from mIMCD-3 cells (left), PC-1 knockout mIMCD-3 cells (middle), and PC-2 knockout mIMCD-3 cells (right). (D) Single channel patch clamp recordings at −100 mV in cell attached configuration for $s$PC-1/PC-2$_{F604P}$ (top) and excised inside-out ciliary patches for mIMCD-3, mIMCD-3 with addition of NTF$^{263\text{-}535}$ in pipette solution, and PC-1 knockout cells (rows 2–4, respectively). (E) Open time histogram of channel openings at −100 mV holding

*Figure 5 continued on next page*

*Figure 5 continued*

potential. (**F**) Time constants of open time distributions in E obtained using one decay equation, \*p<0.05 (n = 7/38 for mIMCD-3 cilia, n = 4/13 for PC-1 KO, n = 4/23 for mIMCD-3 cilia + NTF$^{263-535}$, and n = 6 for sPC-1/PC-2$_{F604P}$). (**G**) Absolute open probabilities for recordings shown in **A**, **B**, and **C**, \*\*p<0.01.

The online version of this article includes the following figure supplement(s) for figure 5:

**Figure supplement 1.** Ciliary recordings in mIMCD-3.

**Figure supplement 2.** Amplitude histograms for excised ciliary patch clamp recordings of mIMCD-3 cilia and PC-1 knockout cilia.

including those carrying the gain of function mutation F604P. More specifically, we show that the CTL domain within the N-terminus of PC-1 is essential for channel activation and that it might function by interacting with the TOP domain of PC-2. In cilia, we provide evidence that an N-terminal fragment containing the CTL domain is sufficient to activate endogenous Ca$^{2+}$-permeable polycystin channels and elevate [Ca$^{2+}$]$_{cilium}$. Thus, we suggest a novel autoregulation mechanism in which the N-terminus of PC-1 activates the channel it is attached to or channels on adjacent cilia following cleavage to become a soluble ligand.

## The PC-1 ectodomain is critical for polycystin complex activation

Our data identify PC-1's CTL domain as an essential component for polycystin activation. Interestingly, the CTL domain is a hotspot for ADPKD-causing mutations (*Dong et al., 2019*), supporting the critical importance of this domain (*Figure 2—figure supplement 1B*). Although C-type lectins are the most diverse family of mammalian carbohydrate-binding proteins (*Keller and Rademacher, 2020*), little is known about the function of the CTL motif within PC-1 (*Sandford et al., 1997*), but it has been reported to bind to carbohydrates in a calcium-dependent manner (*Weston et al., 2001*). All PC-1 family members except PC-1L1 contain an N-terminal CTL, yet PC-1L3 CTL fails to activate the complex. Because the CTL amino acid sequence diverges between different PC-1s, we speculate that each binds to different carbohydrates. It is worth noting that other domains within the N-terminus are also implied in cell-cell recognition, including leucine-rich repeats (LRRs), cell wall integrity and stress response component (WSC), and REJ domain. Future studies addressing the specificity and mechanism of CTL binding, especially to carbohydrate moieties, will be of interest.

Both PC-1 and PC-2 have a large extracellular TOP domain, which contains prominent glycosylation sites. We speculate that the CTL domain in the native PC-1/PC-2 complex interacts with glycans in the TOP domain, which allosterically regulates the ion permeation pathway and/or gating apparatus of the heteromeric polycystin channel (*Vien et al., 2020*; *Figure 7*). In particular, the N-glycan attached to N375 of PC-2 is adjacent to a structure that extends above TM3 and TM4 of the voltage sensor-like domain, so it is conceivable that binding of the CTL domain to this glycan will initiate conformational changes in the voltage sensor-like domain, which could be transmitted to the pore to regulate ion permeation. Indeed, ablation of this glycan by mutation (N375Q) results in a complex that can traffic to the plasma membrane but that remains inactive.

## Plasma membrane-targeted polycystins recapitulate biophysical features of ciliary currents

Signal peptides are cleaved from the nascent polypeptide early in the biosynthetic pathway, thus sPC-1, which is ectopically targeted to the plasma membrane, closely resembles wt PC-1 (*Salehi-Najafabadi et al., 2017*). Likewise, the effective membrane targeting sequence found in the N-terminus of P2Y12 has been successfully used to target adhesion GPCRs to the plasma membrane without compromising functionality (*Liebscher et al., 2015*). The PC-2 F604P mutant has recently been characterized as a gain of function mutant in a *Xenopus* oocyte overexpression model (*Arif Pavel et al., 2016*). It is thought that the F604P substitution within S5 induces conformational changes in the S4-S5 linker (*Zheng et al., 2018*) that lock PC-2 in an open state. Indeed, proline substitutions in the S5 segment of several other TRP family members result in constitutive channel activity (*Grimm et al., 2007*; *Xu et al., 2007*; *Zhou et al., 2017*). A careful comparison of our ciliary recordings of endogenous polycystin channels with membrane-targeted polycystins reveals that both channel populations exhibit comparable biophysical characteristics, lending weight to the effectiveness of our plasma membrane expression system for studying polycystins.

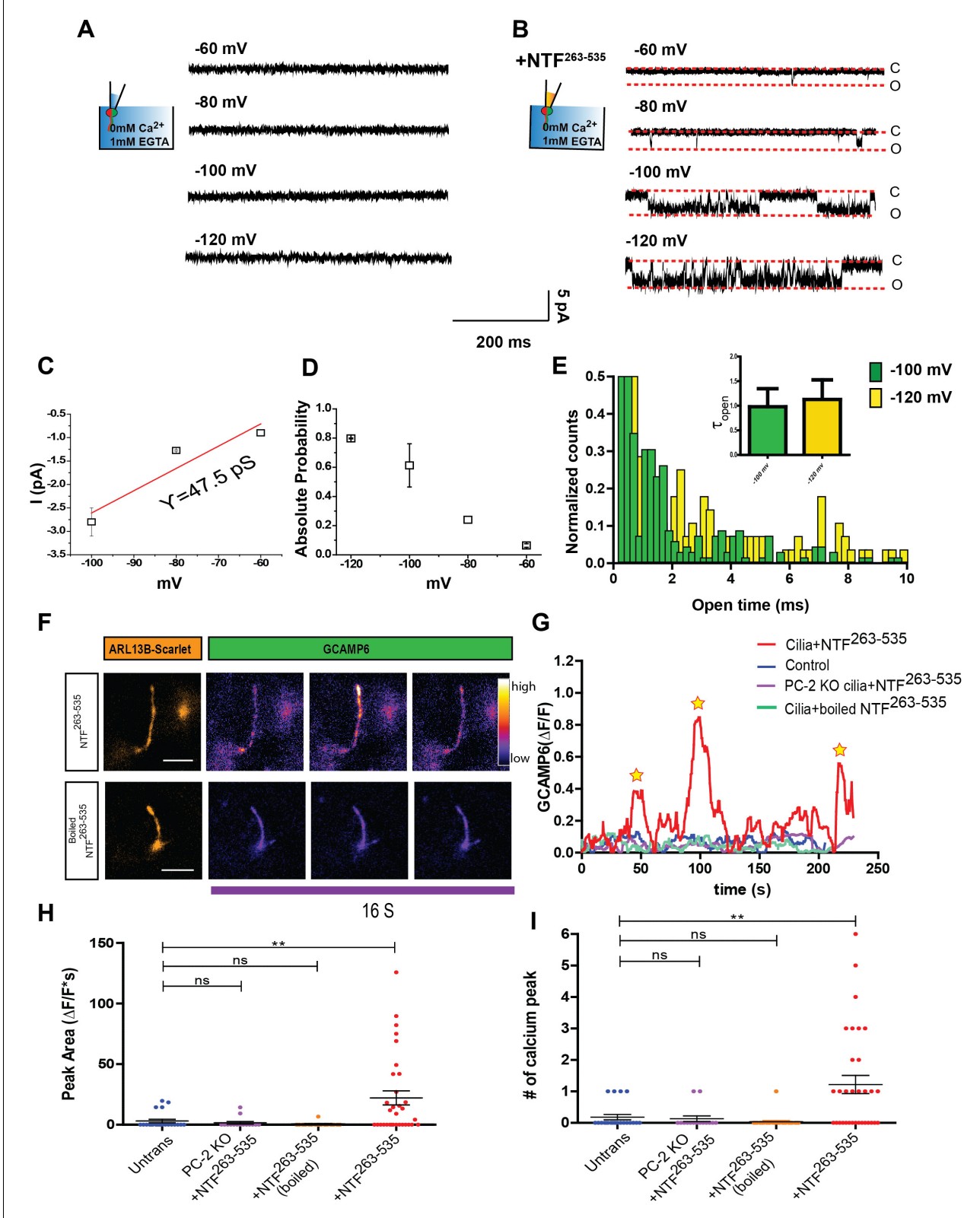

**Figure 6.** NTF[263-535] activates Ca$^{2+}$ influx into primary cilia. (**A**) Excised inside-out patch single channel recordings of mIMCD-3 cilia with 2 mM CaCl$_2$ and 10 mM HEPES in pipette solution at membrane potentials between −60 mV and −120 mV. (**B**) Excised inside-out patch single channel recordings from mIMCD-3 cilia with 0.7 µg/mL of NTF[263-535] in the pipette. Membrane potentials held between −60 mV and −120 mV. (**C**) I-V relationship obtained from ciliary excised patch clamp recordings from mIMCD-3 cilia with NTF[263-535] (n = 3/16). Red line indicates linear fitting. (**D**) Absolute open

*Figure 6 continued on next page*

*Figure 6 continued*
probabilities obtained at membrane potentials between −60 mV and −120 mV (n = 3/16). (**E**) Open time histogram of channel opening events after NTF[263-535] application at −100 and 120 mV. Small bar graph indicates time constants obtained from one decay fits of the histogram (n = 3/16). (**F**) Representative TIRF images after addition of 50 nM NTF[263-535] to mIMCD-3 cells stably expressing Arl13B-scarlet-GCaMP6. GCaMP images are pseudo-colored based on pixel intensity. Scale bar: 5 μm. (**G**) Representative traces of changes in fluorescence (ΔF/F) during TIRFM imaging. Image acquisition every 250 ms. (**H**) Quantification of maximum peak area of fluorescence (ΔF/F*s) for control, PC-2 knock out cells + NTF[263-535], wt mIMCD3 cilia + boiled NTF[263-535], and wt mIMCD3 cilia + NTF[263-535]. (**I**) Quantification of number of calcium spikes observed in TIRFM imaging for control, PC-2 knock out cells + NTF[263-535], wt mIMCD3 cilia + boiled NTF[263-535], and wt mIMCD3 cilia + NTF[263-535]. Quantification is from three independent experiments, **p<0.01.

Our results are in agreement with previous reports of channel activity in the ciliary membrane of renal epithelial cells (*Liu et al., 2018*; *Kleene and Kleene, 2017*; *DeCaen et al., 2013*; *Flannery et al., 2015*). Because no measurable electrical activity remains in mIMCD-3 cells following CRISPR ablation of PC-2, we confirm that the PC-2 subunit is an indispensable component of ciliary currents in kidney epithelial cells (*Liu et al., 2018*; *Kleene and Kleene, 2017*). We also find that primary cilia of mIMCD-3 cells still contain functional channels after CRISPR ablation of PC-1. We speculate that currents recorded from PC-1 knockout cilia are due to homotetrameric PC-2 channels, although we cannot exclude a contribution from other PC-2-containing heteromeric channels such as PC-1L1/PC-2, PC-1L3/PC-2, or PC-2/TRPM3 (*Field et al., 2011*; *DeCaen et al., 2013*; *Flannery et al., 2015*; *Bai et al., 2008*; *Kleene et al., 2019*; *Zhang et al., 2013*). We noted two striking differences between channels in wt and PC-1 knockout mIMCD-3 cilia at voltages close to the resting membrane potential (*DeCaen et al., 2013*). First, the native PC-1/PC-2 heteromeric channel in wt cilia showed very short channel openings compared to the presumably homotetrameric PC-2 channel, supporting the hypothesis raised by cryo-EM data that PC-1 reduces ion flux (*Su et al., 2018*). Second, wt IMCD-3 cilia were electrically silent between membrane potentials of −60 mV and +40 mV whereas channels in PC-1 knockout cilia opened at −40 mV.

## The PC-1 N-terminus likely underlies the molecular mechanism of chemosensation

In our experiments, soluble fragments containing the CTL motif not only restore channel function in polycystin complexes lacking the PC-1 N-terminus but also lengthen the open time duration of endogenous polycystins at negative membrane potentials. This strongly suggests that the PC-1 N-terminus is an agonist for polycystin activation. Our results provide the first insights into why ADPKD-causing mutations are enriched within both CTL and GAIN domains, the latter of which is critical for proteolytic processing and subsequent shedding of the CTL-containing ectodomain. Our results show that soluble fragments of PC-1 potentiate endogenous polycystin channels that already contain one PC-1 N-terminus. Given the predicted three PC-2:1 PC-1 stoichiometry of the heteromeric polycystin complex, we hypothesize that possibly all three PC-2 TOP domains need to be simultaneously engaged by CTL domains in order to trigger a concerted conformational change in their respective voltage sensor-like domains and therefore full channel activation. Alternatively, the tethered PC-1 N-terminus in a native heteromeric polycystin complex may be sterically constrained to participate in intramolecular interactions with its core complex, and thus might engage in intermolecular interactions, either between polycystin complexes on one cilium or as a cilia-cilia proximity sensor.

Alternatively, as-yet-unknown proteins and/or small molecule ligands can bind to modules (such as LRRs) that are in proximity to or in the CTL domain itself. This might mask the carbohydrate-binding site on the CTL domain and thus add another layer of regulation to the polycystin complex. While our results show that CTL is required for channel activation, we cannot exclude the possibility that other more potent agonists of the polycystin complex exist. Future work will determine how the PC-1 N-terminus functions as a polycystin agonist in vivo, either as a soluble ligand or bound to vesicles such as exosomes. We speculate that transport of such ligands by fluids may underline the molecular chemosensory mechanism of primary cilia.

Our data reveal that activation of endogenous polycystins by N-terminal fragments of PC-1 increases ciliary [$Ca^{2+}$] in ~30% of cilia imaged. This is in agreement with previously published work in which only a fraction of cilia shows electrically active polycystin channels. It remains to be

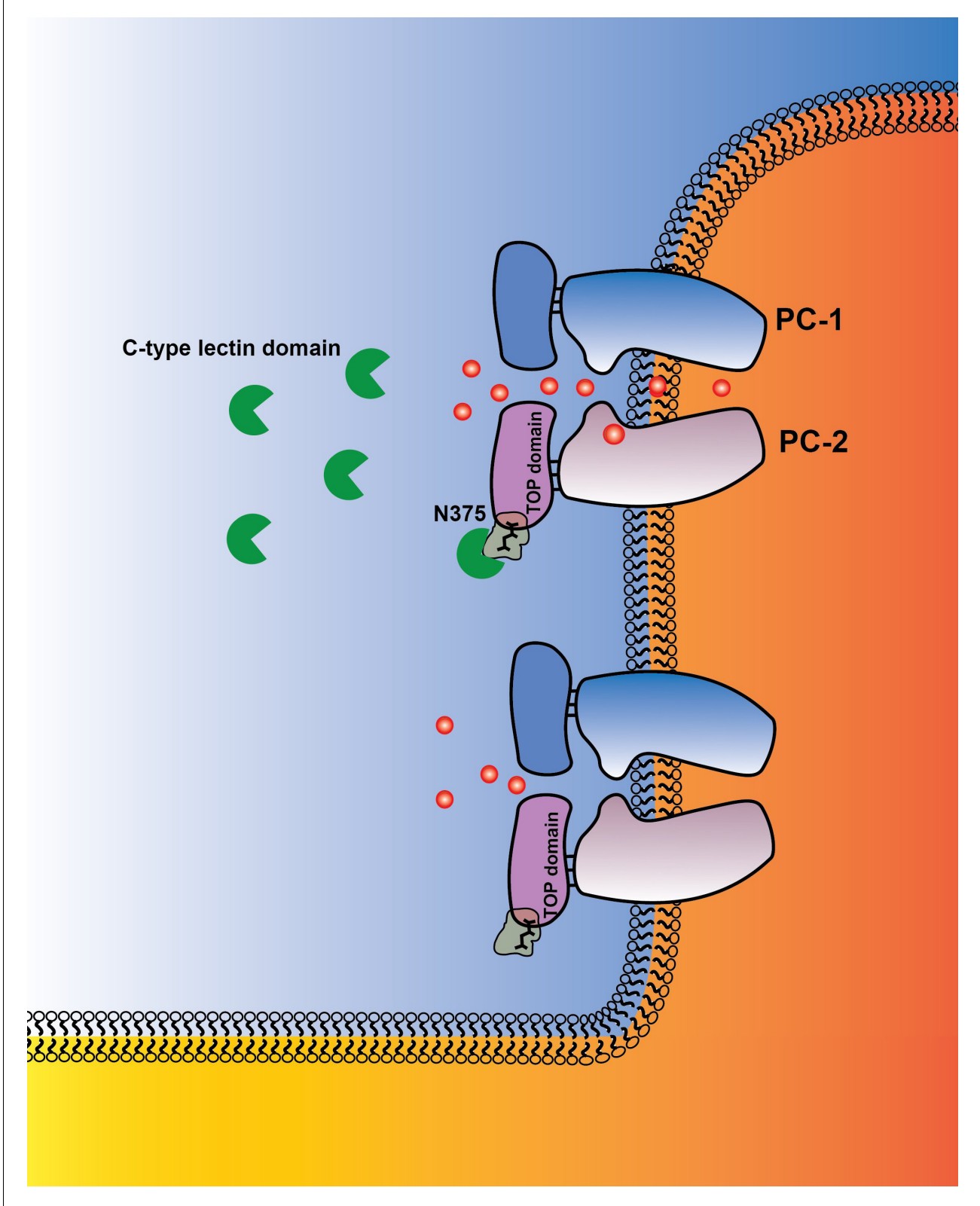

**Figure 7.** Graphical abstract PC-1 N-terminus with its C-type lectin domain interacts with carbohydrates in the PC-2 TOP domain and activates the channel complex in the cilium. PC-1 N-terminus may either activate the channel as a soluble ligand or remain tethered to the channel complex and undergo intermolecular binding to TOP domains on neighboring channels. Channel activity can be different in the presence of N-terminal fragments of PC-1.

determined whether and how agonistic ciliary second messengers, which may only be stochastically present during ciliary recordings, add another layer of regulation to ciliary polycystin channels. Alternatively, there may be an as-yet-unidentified inhibitory signal in cilia that prevents activation of the polycystin complex in some ciliary recordings.

# Materials and methods

## Key resources table

| Reagent type (species) or resource | Designation | Source or reference | Identifiers | Additional information |
|---|---|---|---|---|
| Gene (*Homo sapiens*) | hPKD1 | GenBank | L33243 | |
| Gene (*Homo sapiens*) | hPKD2 | GenBank | NM_000297 | |
| Gene (*Mus musculus*) | mPKD1-L3 | GenBank | NM_001286454 | |
| Gene (*Homo sapiens*) | hArl13b | GenBank | NM_182896 | |
| Cell line (*Mus musculus*) | mIMCD3 | ATCC | CRL-2123 | |
| Cell line (*Mus musculus*) | mIMCD3 PC-1 ko | This paper | Derived from CRL-2123 | CRISPR-CAS9 ablated PC-1 expression |
| Cell line (*Mus musculus*) | mIMCD3 PC-2 ko | From Dr Steven Kleene, University of Cincinnati PMC5283891 | Derived from CRL-2123 | CRISPR-CAS9 ablated PC-2 expression |
| Cell line (*Homo-sapiens*) | HEK293 | ATCC | CRL-1573 | |
| Antibody (mouse) | Anti-FLAG (M2) | Sigma Aldrich | F3165-1MG | (1:100) |
| Antibody (rat) | Anti-HA | Roche | 11867423001 | (1:100) |
| Antibody (rabbit) | Anti-Acetylated tubulin (K40) | Cell signalling | 5335 s | (1:10.000) |
| Antibody (mouse) | Anti-PC-2 (YCE2) | Santa Cruz | YCE2 SC-47734 | (1:1000) |
| Antibody (rat) | Anti-PC-1 (E8) | Baltimore PKD center | E8 | (1 µg/mL western blot) |
| Recombinant DNA reagent | pTRE3G-Bi | Takara Bio | Cat # 631337 | |
| Recombinant protein | PC-1 NTF 24–852 | This paper | Amplified from L33243 | |
| Recombinant protein | PC-1 NTF 263–535 | This paper | Amplified from L33243 | |
| Software, algorithm | PRISM10 | Graphpad | | |
| Software, algorithm | pCLAMP | Molecular Devices | | |
| Software, algorithm | Clampfit10.6 | Molecular Devices | | |
| Other | Höchst 33342 | Thermo Fisher | | 1:10.000 |

## Molecular biology

hArl13B-EGFP was described previously. Arl13b-scarlet-GCaMP6s is an updated version of our previously characterized ciliary $Ca^{2+}$ sensor Arl13b-mCherry-GECO1.2 in which mcherry and GECO1.2 have been replaced with mScarlet and GCaMP6s, respectively. κ IgG HA-PC-1L3 (sPC-1L3) has been described previously (*Salehi-Najafabadi et al., 2017*). sPC-1 was generated by using a unique BgllI site within hPKD1 to replace the endogenous signal peptide with the 12 aa k IgG leader sequence followed by HA tag using PCR amplification and Gibson assembly (NEB). $^{HA}$PC-1 was generated by inserting an HA tag between the predicted endogenous signal peptide and mature polypeptide chain following the same protocol. For the construction of FLAG-PC-2, several positions within the extracellular domains of PC-2 with negatively charged amino acids were tested. We identified one region at position 372 that tolerated the insertion of the FLAG sequence, based on surface trafficking. Chimera constructs of PC-1 and PC-1L3 were generated by substituting the ectodomain at the GPS cleavage site with the corresponding orthologue using PCR and Gibson assembly. All PC-1 and PC-2 combinations were cloned into pTRE3G-Bi vector (Takara Bio). The MCSI site was used for PC-1 variants while the MCSII site was used for PC-2 variants. This ensured simultaneous translation of PC-1 and PC-2 from the same plasmid. All DNA sequences were confirmed by sequencing (ELIM Bio).

## Antibodies

Rabbit anti-acetylated tubulin (K40)(D20G3) Cell Signaling Technology (5335 s); rat anti-hemaglutinin (HA), Roche (11867423001); mouse anti-FLAG M2, Sigma-Aldrich (F3165-1MG); rat anti PC1 antibody (E8) (Baltimore PKD core center); mouse anti PC2 (YCE2) antibody, Santa Cruz Biotechnology (SC-47734).

## Immunocytochemistry and confocal microscopy

Cells were fixed with 4% formaldehyde, permeabilized with 0.2% Triton X-100, and blocked by 2% FBS, 2% BSA, and 0.2% fish gelatin in PBS. Cells were labeled with the indicated antibody and secondary goat anti-rabbit, anti-rat or anti-mouse fluorescently labeled IgG (Thermo Fisher) and Hoechst 33342 (Thermo Fisher). Confocal images were obtained using a Nikon spinning disk with a 63× oil immersion, 1.2 N.A. objective, or a 100× oil immersion, 1.4 N.A. objective at the UCSF Nikon Imaging Core. Images were further processed using ImageJ (NIH).

## Cell culture and quantification of plasma membrane PC-1 and PC-2

Tet inducible mIMCD3 and HEK293 cells were generated from parental lines (obtained from ATCC) according to the manufacture's protocol (Takara Bio). HEK or IMCD3 cells expressing the tet activator were transiently co-transfected with the indicated pTRE3G-Bi vector (Takara Bio) and the EGFP vector using Lipofectamine LTX for mIMCD3 or Lipofectamine2000 for HEK293 (Thermo Fisher). Surface trafficking of PC-1 and PC-2 was measured by quantifying the amount of anti-HA or anti-FLAG antibody bound to the plasma membrane, respectively. 24 hr after transfection 1 ug/mL doxycycline was added and cells were incubated for an additional 24 hr. For surface staining, adherent live cells were incubated with opti-MEM containing either anti-HA antibody (1:100) or anti-FLAG antibody (1:100) for 20 min at room temperature to avoid internalization of antibodies. Cells were washed twice with opti-MEM and fixed with 4% PFA. After incubation with blocking buffer, cells were labeled with secondary goat anti-rat or goat anti-mouse antibodies conjugated to Alexa 647. Several Z stacks of indicated groups were acquired using identical settings and 647 fluorescence was quantified on all GFP positive cells using ImageJ. This approach allowed an unbiased quantification of membrane inserted HA. In addition, expression of the proteins was confirmed in total staining using permeabilized cells. mIMCD3 cells with ablated PC-2 expression were described previously (*Kleene and Kleene, 2017*). Cells were tested for mycoplasma contamination regularly (Lonza). Data are representative of at least three independent experiments.

## Electrophysiology

Recordings were performed using a multiclamp 200B (Axon Instruments), digitized using a digidata 1324A (Axon Instruments) and recorded using pClamp software (Axon Instruments). Whole cell configuration patch clamp data were filtered at 1 kHz and sampled at 10 kHz. Unless stated otherwise,

the voltage step pulse was applied from −100 mV to 180 mV in 20 mV increments during 150 ms and holding potential was given at −60 mV. For the voltage ramp pulse, the same range of voltage steps was applied with 500 ms duration. The resistances of pipettes for whole cell and ciliary patch clamp were 6–8 MΩ and 18–24 MΩ, respectively. The tip of the pipette was further polished using Narishige MF-830 microforge equipped with a 100× Nikon objective. For patch clamp experiment, the extracellular solution consisted of (mM): 145 Na-gluconate, 5 KCl, 2 $CaCl_2$, 5 $MgCl_2$, 10 HEPES, and adjusted to pH 7.4 using NaOH. The intracellular solution was used as follows: 90 NaMES, 10 NaCl, 2 $MgCl_2$, 10 HEPES, 5 EGTA, 100 nM free calcium adjusted by $CaCl_2$, and adjusted to pH 7.4 using NaOH. The free calcium was calculated using CaBuf software (G.Droogmans, Leuven, Belgium). For calcium permeability test, the extracellular solution consisted of (mM): 2 mM $CaCl_2$ and 10 HEPES adjusted to pH 7.4 using Trizma base and adjusted to 295mOsm using mannitol. The intracellular solution consisted of (mM): 10 HEPES and 1 EGTA adjusted to pH 7.4 Trizma base.

## Statistical and data analysis for electrophysiology

Ciliary excised patch clamp data and whole cell configuration patch clamp were analyzed with Clampfit10.6 (Axon Instruments/Molecular Devices), Origin8 (Originlab), and Prism10.0 (Graphpad). Data are shown as mean ± SEM, and $n$ represents independent experiments for the number of tested cells in electrophysiology. For relative open probability of $s$PC-1-PC-2$_{F604P}$, the data obtained at −80 mV tail pulse were fitted to a Boltzmann distribution using Origin8 (Originlab).

$P_o(V)$ = $P_{-100}$ +($P_{+180}$−$P_{-100}$ )/(1+exp[($V_{1/2}$ −V)/$\kappa$]) where $P_{-100}$ and $P_{+180}$ are the open probabilities of the channel at the most negative potential (−100 mV) and the most positive potential (+180 mV), respectively. V indicates the membrane potential, $V_{1/2}$ is the half-maximal activation potential, and $\kappa$ is the slope factor. For single channel open probability, $P_{open}$ was calculated by:

$$Popen = \frac{to}{T},$$

where the total time that the channel presented in the open state and T is the total observation time. If a patch contains more than one of the same type of channel, Popen was computed by:

$$Popen = \frac{to}{NT},$$

where, N indicates the number of channels in the patch. We used the following equation to populate data.

$$To = \sum Lto,$$

where, L indicates the level of the channel opening. The absolute probability of the channel being open NPo is computed by:

$$NPo = \frac{To}{To + Tc},$$

where, Tc indicates the total closed time

## $Ca^{2+}$-imaging in primary cilia

IMCD3 cells expressing Arl13b-mScarlet-GCaMP6 in primary cilia were imaged as described previously (*Ishikawa and Marshall, 2015*). In brief, cells were grown in the bottom side of 24 mm transwell insert with 8 um mesh size allowing rapid exchange of fluids across the membrane. Cilia were observed under an inverted Nikon microscope equipped with an TIRF imaging setup. Images were acquired 1 min after applying 1 μg/mL PC-1 NTF into the transwell. This assay allowed fast imaging along the entire length of the cilium. In most cases, cilia were localized by 561 nm (mScarlet) excitation and imaged in 488 nm (GCaMP6) and using the full CCD chip (200 ms exposure; five fps). Fluorescence was quantified and processed using ImageJ and Python.

## Expression of N-terminal fragments of PC-1

Constructs were designed to produce various PC-1 N-terminal fragments as secreted proteins in HEK293S GnT1$^{-/-}$ cells using the BacMam expression system. In brief, the endogenous signal

sequence of PC-1 was replaced with the strong IgG kappa leader peptide followed by either a M1-FLAG epitope or a maltose-binding protein fusion that allows for affinity purification using anti-M1 Flag antibody or amylose resin, respectively. HEK293S GnT1-/- cells were transduced with baculoviruses when cell density reaches $1 - 2 \times 10^6$ cells/mL. Sodium butyrate was added to at a final concentration of 5 mM to enhance protein expression 8–12 hr post-transduction. Temperature was reduced to 30°C and the supernatant was harvested 5 days post-transduction. Various PC-1 N-terminal fragments were purified by affinity chromatography whereby secreted PC-1 fragments were captured by passing supernatant through affinity resins using gravity. The purified PC-1 fragments were then exchanged to a buffer containing 20 mM HEPES, 150 mM NaCl, pH 7.4 via extensive dialysis at 4C. PC-1 N-terminal fragments were then flash-frozen in liquid nitrogen and stored as aliquots at −80°C until use.

### Data analysis

Group data are presented as mean ± SEM. Statistical comparisons were made using unpaired student t-tests for electrophysiology and quantification of surface expression. The Mann-Whitney U-test was used for comparing peak area of $Ca^{2+}$ signals and Fisher's exact test was used for comparing # of $Ca^{2+}$ peaks (Prism). Statistical significance is denoted with an asterisk (*p<0.05; **p<0.01).

## Acknowledgements

This work was supported by National Institute of Health Grant R01GM130908 (MD), R01 DK111611 (FQ), Polycystic Kidney Disease Research Resource Consortium U54DK126114 (FQ), the Fritz Thyssen Foundation (MD), the National Research Foundation of Korea (NTF) grant funded by the Korean government (MSIT) (No.2019R1A6A3A03033302) (KH), PKD Foundation (236G19a) (FQ) and a fellowship grant (215F19a) (RW). EC has been supported by NIH grant R01 DK110575 and is a Pew scholar supported by the Pew Charitable Trusts.

We thank John King for help with the analysis of single channel recordings and Yuriy Kirichok for advice on electrophysiology. We are grateful to Julia Doerner, David Clapham, Ray Hulse, and Anthony Marantos for their critical reading of the manuscript and helpful discussions.

## Additional information

### Funding

| Funder | Grant reference number | Author |
|---|---|---|
| National Institute of General Medical Sciences | R01GM130908 | Markus Delling |
| Fritz Thyssen Stiftung | | Markus Delling |
| National Research Foundation of Korea | 2019R1A6A3A03033302 | Kotdaji Ha |
| National Institute of Diabetes and Digestive and Kidney Diseases | R01 DK125404 | Feng Qian |
| National Institute of Diabetes and Digestive and Kidney Diseases | U54DK126114 | Feng Qian |
| National Institute of Diabetes and Digestive and Kidney Diseases | DK110575 | Erhu Cao |
| PKD Foundation | 236G19a | Feng Qian |
| PKD Foundation | 215F19a | Rebecca V Walker |
| National Institute of Diabetes and Digestive and Kidney Diseases | R01 DK111611 | Feng Qian |

The funders had no role in study design, data collection and interpretation, or the decision to submit the work for publication.

## Author contributions
Kotdaji Ha, Conceptualization, Formal analysis, Validation, Investigation, Visualization, Methodology, Writing - original draft, Project administration, Writing - review and editing; Mai Nobuhara, Formal analysis, Validation, Investigation, Visualization, Methodology, Project administration; Qinzhe Wang, Resources, Formal analysis, Investigation, Visualization, Methodology; Rebecca V Walker, Resources, Investigation; Feng Qian, Resources, Supervision, Funding acquisition, Writing - original draft; Christoph Schartner, Resources, Visualization, Methodology; Erhu Cao, Conceptualization, Resources, Formal analysis, Supervision, Funding acquisition, Methodology, Writing - original draft, Writing - review and editing; Markus Delling, Conceptualization, Resources, Formal analysis, Supervision, Funding acquisition, Investigation, Visualization, Methodology, Writing - original draft, Writing - review and editing

## Author ORCIDs
Rebecca V Walker (iD) http://orcid.org/0000-0002-2473-4303
Markus Delling (iD) https://orcid.org/0000-0001-9556-2097

## Decision letter and Author response
Decision letter https://doi.org/10.7554/eLife.60684.sa1
Author response https://doi.org/10.7554/eLife.60684.sa2

# Additional files

## Supplementary files
• Transparent reporting form

## Data availability
Previously published data from PDB was used, available under the accession code 6A70. All data generated or analysed during this study are included in the manuscript and supporting files.

The following previously published dataset was used:

| Author(s) | Year | Dataset title | Dataset URL | Database and Identifier |
|---|---|---|---|---|
| Su Q, Hu F, Ge X, Lei J, Yu S, Wang T, Zhou Q, Mei C, Shi Y | 2018 | Structure of the human PKD1/PKD2 complex | https://www.rcsb.org/structure/6A70 | RCSB Protein Data Bank, 6A70 |

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
