## [Decision Letter]

**Acceptance summary:**

The polycystin complex, PKD1/PC-1 and PKD2/ PC-2, forms a cation channel in the primary cilia, but the gating and permeation properties were controversial, largely due to the technical difficulty in recording from ciliary membranes. In this manuscript by Ha et al., the authors have successfully, through various molecular manipulations, re-routed the PC1/PC2 protein complex to the plasma membrane of mammalian cells. It was demonstrated that the N-terminal extracellular domain (NTF) of PC1, when proteolytically cleaved, may serve as an endogenous agonist of the heteromeric PC1/PC2 channel. The results have provided another platform for studying the regulation of PC1/PC2 by endogenous cellular cues acting on the primary cilia. Additionally, it has made it possible to conduct high-throughput screening of small-molecule PC1/PC2 modulators, which can reveal novel therapeutic approaches to treat ciliary diseases.

**Decision letter after peer review:**

Thank you for submitting your article "The heteromeric PC-1/PC-2 polycystin complex is activated by the PC-1 N-terminus" for consideration by *eLife*. Your article has been reviewed by three peer reviewers, and the evaluation has been overseen by a Reviewing Editor and Richard Aldrich as the Senior Editor. The following individual involved in review of your submission has agreed to reveal their identity: Haoxing Xu (Reviewer #1).

The reviewers have discussed the reviews with one another and the Reviewing Editor has drafted this decision to help you prepare a revised submission.

Summary:

In this manuscript, Ha et al., report a successful reconstitution of the ciliary polycystin PC-1/PC-2 complex in mammalian cells. They show that the complex forms a cation channel that mediates an outwardly rectifying ion current in the plasma membrane. Moreover, this study demonstrates that soluble N-terminal extracellular domain of PC-1 can activate the channel and act as an endogenous agonist. These findings are corroborated by state-of-the-art electrophysiological recording and calcium imaging of cilia. The results are significant, because they have provided another platform for studying the regulation of PC1/PC2 by endogenous cellular cues acting on the primary cilia. Additionally, it has made it possible to conduct high-throughput screening of small-molecule PC1/PC2 agonists and inhibitors, which may provide not only some studying tools, but also possible therapeutic approaches to treat autosomal dominant polycystic kidney disease (ADPKD). Taken together, the research provides strong evidence for functional PC-1/PC-2 complex and novel mechanistic insight to understand mechanism of ADPKD.

1) There is some background, outwardly rectifying TRPM7-like currents in most mammalian cell lines, including HEK293 cells. Hence, it is less convincing to compare the results from different recordings/cells under different experimental conditions, even if blind experiments are conducted. Have authors tried bath-application of NTF peptides in the whole-cell recordings of sPC-1/PC2 or sPC-1/PC2-_F604P_-expressing cells, to see whether you can see the current activation and washout in the same cells?

2) In Figure 5F, authors may consider to boost the Ca^2+^ signal by increasing the extracellular Ca^2+^ concentration. Have authors tried bath-application of NTF peptides to PC1/PC2 overexpressing mIMCD3 cells in the GCaMP imaging experiments?

3) One major point that needs to be addressed is the labeling of the different constructs and how they are cited in the text/legends or the figures. This creates major confusion and needs to be carefully checked throughout the whole manuscript. See the specifics below.

4) Figure 1A: PC-1 should be sPC-1?

5) Figure 1B, C, D: second panel – PC-1 is HA-PC-1?

6) Figure 2B: sPC is missing for all panels – please make sure that all figures are labeled with the same construct names – the labeling seems to be all over the place (also see below).

7) Figure 2D: the constructs again have different names, which are not explained in the legend or the text.

8) Figure 2C, please double check whether there is no error in scale bar and unify them for easy comparison. Weak currents are elicited in PC-1L3 and PC-11L3NT. These are different with residual current from PC-1^ΔNT^ and PC-11L3CTL. PC-1^ΔNT^ seems to be also responsive to voltage change because it has tail currents, indicating potential changes in channel conformation. If so, loss of N-terminus of PC-1 does not seem to alter voltage sensing of the polycystin complex.

9) Not all claims in the text are justified by the data shown in the figures.

10) Subsection “PC-1 and PC-2 form functional channels in the plasma membrane”, first paragraph: Figure 1—figure supplement 1E does not show sPC-1 and/or sPC-2 expression – please show images for IMCD-3 cells.

11) Figure 1—figure supplement 1B: labeling against HA for PC-1 is missing. For quantification, a ratio of the extracellular HA and flag signal should be calculated, demonstrating that they are shuttled simultaneously into the PM (subsection “PC-1 and PC-2 form functional channels in the plasma membrane”, first paragraph).

12) Subsection “PC-1 and PC-2 form functional channels in the plasma membrane”, second paragraph: while the authors' conclusion that PC-1 contributes to the pore of the channel complex and PC-1/PC-2 subunits form a functional channel is valid, it is not clear how the open probability is calculated here. Figure 1K does not show channel open probability (it is a normalized I/V curve) – I/Imax cannot serve as Po. GV curve is commonly used to present channel conductance. Besides, they are normalized by their own Imax, thus the curve is not comparable to each other. It is also unusual that the authors try to obtain open probability at negative command voltage, not positive, for the outwardly rectifying channel?

13) Subsection “The PC-1 N-terminus is essential for polycystin complex activation”, first paragraph: Figure 2B does not provide any data for PC-2_F604P_ transfected cells – are the panels and the legend correct? If yes, then the data for the mutant needs to be shown.

14) No images analyzing the plasma membrane localization are shown for PC-1L3/1CTL – please provide the data.

15) Current traces for PC-1^ΔNT^;PC-2_F604P_ are missing in Figure 2 – please provide the data.

16) Subsection “The PC-1 N-terminus is essential for polycystin complex activation”, last paragraph: pore mutant is not show in Figure 2F, at least the legend does not indicate that this construct has been used. Please comment or include the data set.

17) Subsection “The PC-1 N-terminus is essential for polycystin complex activation”, last paragraph: where is the condition PC-1L2^s1NT^;PC-2_F604P_? The traces in Figure 2C do not show this data set and the labeling in 2C and D is different.

18) Subsection “Polycystin activation depends on CTL and TOP domains”, first paragraph, please indicate the voltage. Also please use the protein names in the text consistent with the labels in Figures (Figure 2C). Is signal pipette replaced in PC-1 used here?

19) Subsection “Polycystin activation depends on CTL and TOP domains”, second paragraph: when comparing the data in Figure 3B to the data in Figure 1C, there seems to be a difference (opposite to what is written in the text) – please perform statistical analysis between the two conditions.

20) Missing information about the concentration of the purified NT domains – how much was added for the different measurements? Has the protein been added in excess or in stochiometric amounts?

21) Some ephys data are loosely formatted to support the authors' conclusion, which the authors need to consider to better re-present by more careful analysis and/or text revision.

22) Figure 5. Here, the authors also obtained open probability in panel D. Is it calculated in the same way as Figure 1 or from single channel conductance?

Revisions expected in follow-up work:

As mentioned in ("Revisions for this paper" section #1) similar experiments could also be performed in the whole-cilium (or out-side-out) recordings. The significance of the study can be further boosted if you can see PC-1/PC-2 current activation by a protease cocktail, e.g., trypsin. However, such experiments are not required for the revision.

---

## [Author Response]

Revisions for this paper:1) There is some background, outwardly rectifying TRPM7-like currents in most mammalian cell lines, including HEK293 cells. Hence, it is less convincing to compare the results from different recordings/cells under different experimental conditions, even if blind experiments are conducted. Have authors tried bath-application of NTF peptides in the whole-cell recordings of sPC-1/PC2 or sPC-1/PC2-_F604P_-expressing cells, to see whether you can see the current activation and washout in the same cells?

We agree with the reviewer that endogenous currents, especially TRPM7-mediated currents, can easily be misinterpreted as currents generated by overexpressed channel. Hence we have taken great care to eliminate the contribution of TRPM7: In order to block endogenous TRPM7 currents we added 5 mM MgCl_2_ to the extracellular and 1 mM MgCl_2_ to the intracellular solution. In none of our recordings did we ever observe a “run-up” of currents over time, a hallmark of TRPM7-mediated currents. In the revised manuscript we now include data showing that acute perfusion of sPC-1^ΔNT^/PC-2_F604P_ expressing cells with PC1-NTF potentiates polycystin currents in whole-cell recordings.

This data is now included in Figure 3D. Further, as suggested by the reviewers, we also tested whether trypsin digestion of PC-1 N-terminus reduces polycystin currents. As shown in Figure 3H, acute perfusion with trypsin inhibits polycystin dependent currents, likely by removing PC-1 N-terminus from sPC-1/PC-2_F604P_ expressing cells. These findings are further supporting our hypothesis that PC-1 N-terminus and C-type lectin domain are critical for channel activity.

2) In Figure 5F, authors may consider to boost the Ca^2+^ signal by increasing the extracellular Ca^2+^ concentration. Have authors tried bath-application of NTF peptides to PC1/PC2 overexpressing mIMCD3 cells in the GCaMP imaging experiments?

We agree with the reviewer that acute perfusion of primary cilia with PC-1 NTF would be ideal. However we have been using a TIRF imaging system at the Nikon Imaging Core, which is not set up for perfusion. Due to the reduced staffing of the core we have no possibility to upgrade the system anytime soon. Further, as described in the Materials and methods section, TIRF imaging modality requires us to image cilia on cells grown on inverted transwell dishes.

3) One major point that needs to be addressed is the labeling of the different constructs and how they are cited in the text/legends or the figures. This creates major confusion and needs to be carefully checked throughout the whole manuscript. See the specifics below.

Following the reviewers’ suggestion we have now included a table (Table 1) with a detailed description of all constructs, including the exact amino acid position for chimeras or insertion (14 constructs in total, four of which address chimeras between PC-1 and PC-1L3).

4) Figure 1A: PC-1 should be sPC-1?

We corrected Figure 1A accordingly.

5) Figure 1B, C, D: second panel – PC-1 is HA-PC-1?

We corrected Figure 1B, C, D accordingly.

6) Figure 2B: sPC is missing for all panels – please make sure that all figures are labeled with the same construct names – the labeling seems to be all over the place (also see below).

That you for pointing this out. We unified labeling across all figures, including panels in Figure 2B.

7) Figure 2D: the constructs again have different names, which are not explained in the legend or the text.

We unified labeling of the constructs and now refer to a detailed description of all constructs in Table 1 (see above).

8) Figure 2C, please double check whether there is no error in scale bar and unify them for easy comparison. Weak currents are elicited in PC-1L3 and PC-11L3NT. These are different with residual current from PC-1^ΔNT^ and PC-11L3CTL. PC-1^ΔNT^ seems to be also responsive to voltage change because it has tail currents, indicating potential changes in channel conformation. If so, loss of N-terminus of PC-1 does not seem to alter voltage sensing of the polycystin complex.

Thank you for noticing this apparent inconsistency. We have now unified the scale for each recording. Upon close examination we realize that the apparent tail currents from PC-1^ΔNT^F+PC2 _F604P_ recordings were not a direct consequence of channel expression. We have now addressed this issue and show that the PC-1^ΔNT^F+PC2 _F604P_ construct does not show apparent voltage sensitivity.

9) Not all claims in the text are justified by the data shown in the figures.

We carefully adjusted all claims to the data presented in the manuscript. We also modified the Discussion accordingly.

10) Subsection “PC-1 and PC-2 form functional channels in the plasma membrane”, first paragraph: Figure 1—figure supplement 1E does not show sPC-1 and/or sPC-2 expression – please show images for IMCD-3 cells.

Figure 1—figure supplement 1E showed the ciliary localization of PC-1^ΔNT^ and PC-2FLAG using surface HA or surface FLAG staining. This was to emphasize the point that in ciliated cells the proteins still enrich in their native environment. We now included one set of images showing that sPC1 also enriches in the cilium, when present.

11) Figure 1—figure supplement 1B: labeling against HA for PC-1 is missing. For quantification, a ratio of the extracellular HA and flag signal should be calculated, demonstrating that they are shuttled simultaneously into the PM (subsection “PC-1 and PC-2 form functional channels in the plasma membrane”, first paragraph).

Over the course of many experiments we noted that simultaneous labeling with anti HA and anti FLAG antibodies results in varying and unpredictable degrees of surface staining for HA or FLAG. We currently hypothesize that due to steric hindrance both antibodies may exclude each other from binding to the extracellular tags. We realized that quantification of surface HA or surface FLAG staining, which has been processed separately but is from the same set of transfected cells, is the most reliable approach. We have added a set of images for surface cilia surface staining of sPC-1/PC-2 expressing IMCD3 cells, see above

12) Subsection “PC-1 and PC-2 form functional channels in the plasma membrane”, second paragraph: while the authors' conclusion that PC-1 contributes to the pore of the channel complex and PC-1/PC-2 subunits form a functional channel is valid, it is not clear how the open probability is calculated here. Figure 1K does not show channel open probability (it is a normalized I/V curve) – I/Imax cannot serve as Po. GV curve is commonly used to present channel conductance. Besides, they are normalized by their own Imax, thus the curve is not comparable to each other. It is also unusual that the authors try to obtain open probability at negative command voltage, not positive, for the outwardly rectifying channel?

We are sorry for the confusion. We agree with the reviewer that the labeling of the figure was not precise. Here we are comparing the relative open probability between both channels and calculated the relative open probability based on inward currents at the tail pulse -80mV for each of the voltage steps. Further we now include statistical analysis at representative voltages.

13) Subsection “The PC-1 N-terminus is essential for polycystin complex activation”, first paragraph: Figure 2B does not provide any data for PC-2_F604P_ transfected cells – are the panels and the legend correct? If yes, then the data for the mutant needs to be shown.

We included sPC-1/PC-2 _F604P_ data in Figures 1B and C. In none of the experiments did we notice a major difference in surface trafficking between wtPC-2 and the F604P mutant, see also Figure 1C. We thus used PC-2 for quantification.

14) No images analyzing the plasma membrane localization are shown for PC-1L3/1CTL – please provide the data.

Imaging analysis data for PC-1L3/1CTL is included in Figure 2D.

15) Current traces for PC-1^ΔNT^;PC-2_F604P_ are missing in Figure 2 – please provide the data.

Current traces for PC-1^ΔNT^ and PC-2_F604P_ has been included in Figure 2C.

16) Subsection “The PC-1 N-terminus is essential for polycystin complex activation”, last paragraph: pore mutant is not show in Figure 2F, at least the legend does not indicate that this construct has been used. Please comment or include the data set.

The pore mutant for sPC-1/PC-2_F604P_ data set was already shown in the Figure 1K. We now updated the figure legend.

17) Subsection “The PC-1 N-terminus is essential for polycystin complex activation”, last paragraph: where is the condition PC-1L2^s1NT^;PC-2_F604P_? The traces in Figure 2C do not show this data set and the labeling in 2C and D is different.

We have not included any data for PC-1L2. As of now we could not obtain reliable HA surface staining with sPC-1L2 and PC-2. We also did not measure any macroscopic currents with sPC-1L2+PC-2_F604P_.

18) Subsection “Polycystin activation depends on CTL and TOP domains”, first paragraph, please indicate the voltage. Also please use the protein names in the text consistent with the labels in figures (Figure 2C). Is signal pipette replaced in PC-1 used here?

We now mention the voltage pulse (+100 mV). As discussed above we also updated the name of the constructs used in this study, see Table 1. And yes, the construct contains the secretion sequence (sPC-1).

19) Subsection “Polycystin activation depends on CTL and TOP domains”, second paragraph: when comparing the data in Figure 3B to the data in Figure 1C, there seems to be a difference (opposite to what is written in the text) – please perform statistical analysis between the two conditions.

The reviewer noticed the difference in y axis, thank you very much. The initial discrepancy in values resulted from the fact that the images were taken from different experiments and different microscopes. We now provide data in which we analyzed surface expression of sPC-1/PC-2 and the N375Q mutant under identical settings and performed statistical analysis.

20) Missing information about the concentration of the purified NT domains – how much was added for the different measurements? Has the protein been added in excess or in stochiometric amounts?

We had mentioned the concentration of the purified NTF domain in the subsection “N-terminal PC-1 fragments activate ciliary polycystin complexes”. We used 0.7 μg/mL (equal to ~50 nM final concentration) of NTF263-535 in the recording pipette and the same amount of the protein was applied for the calcium imaging.

21) Some ephys data are loosely formatted to support the authors' conclusion, which the authors need to consider to better re-present by more careful analysis and/or text revision.

We are sorry that the reviewer was under this impression. We had purposely omitted details in the manuscript in order to improve readability. As already discussed above we have now reformatted our figures for a more homogenous presentation.

22) Figure 5. Here, the authors also obtained open probability in panel D. Is it calculated in the same way as Figure 1 or from single channel conductance?

The open probability presented in the Figure 5D was obtained using single channel conductance.

Popen was computed by:

Popen=toT where to indicates the total time for the channel in the open state and T indicates the total observation time. This equation applied only for a single channel in the patch. If a patch contains multiple channel with the same type, we applied the equation below: Popen=toNT where N is the number of channels in the patch, and: To=∑Lto where L is the level of the channel opening. The absolute probability of the channel being open NPo is computed by: NPo=ToTo+Tc where Tc indicates the total closed time. We have now added a detailed description of how we obtained open probability in the Materials and methods section.

Revisions expected in follow-up work:As mentioned in ("Revisions for this paper" section #1) similar experiments could also be performed in the whole-cilium (or out-side-out) recordings. The significance of the study can be further boosted if you can see PC-1/PC-2 current activation by a protease cocktail, e.g., trypsin. However, such experiments are not required for the revision.

We followed the reviewer’s suggestion and perfused HEK cells expressing sPC-1/PC-2_F604P_ with 0.125% trypsin This test resulted in the significant reduction of the currents (8.9 ± 1.4 pA/pF,) n=6 from sPC-1/PC-2_F604P_ (52.0 ± 17.1 pA/pF, n=6) by 78.4 ± 4.9 %. In contrast, untransfected cells did not show any significant decrease after trypsin application. We suggest that trypsin addition digested the PC-1 N-terminus and thus ultimately behaves like PC-1^ΔNT^. We added these results in the revised Figure 3H-K.